# A molecular timescale for eukaryote evolution with implications for the origin of red algal-derived plastids

Jürgen F. H. Strassert [1,5,7], Iker Irisarri [1,2,6,7], Tom A. Williams [3] & Fabien Burki [1,4✉]

In modern oceans, eukaryotic phytoplankton is dominated by lineages with red algal-derived plastids such as diatoms, dinoflagellates, and coccolithophores. Despite the ecological importance of these groups and many others representing a huge diversity of forms and lifestyles, we still lack a comprehensive understanding of their evolution and how they obtained their plastids. New hypotheses have emerged to explain the acquisition of red algal-derived plastids by serial endosymbiosis, but the chronology of these putative independent plastid acquisitions remains untested. Here, we establish a timeframe for the origin of red algal-derived plastids under scenarios of serial endosymbiosis, using Bayesian molecular clock analyses applied on a phylogenomic dataset with broad sampling of eukaryote diversity. We find that the hypotheses of serial endosymbiosis are chronologically possible, as the stem lineages of all red plastid-containing groups overlap in time. This period in the Meso- and Neoproterozoic Eras set the stage for the later expansion to dominance of red algal-derived primary production in the contemporary oceans, which profoundly altered the global geochemical and ecological conditions of the Earth.

[1] Department of Organismal Biology, Program in Systematic Biology, Uppsala University, Uppsala, Sweden. [2] Department of Biodiversity and Evolutionary Biology, Museo Nacional de Ciencias Naturales (MNCN-CSIC), Madrid, Spain. [3] School of Biological Sciences, University of Bristol, Life Sciences Building, Bristol, UK. [4] Science for Life Laboratory, Uppsala University, Uppsala, Sweden. [5] Present address: Department of Ecosystem Research, Leibniz Institute of Freshwater Ecology and Inland Fisheries, Berlin, Germany. [6] Present address: Department of Applied Bioinformatics, Institute for Microbiology and Genetics, University of Göttingen, and Campus Institute Data Science (CIDAS), Göttingen, Germany. [7] These authors contributed equally: Jürgen F. H. Strassert, Iker Irisarri. ✉email: fabien.burki@ebc.uu.se

Plastids (e.g. chloroplasts) are organelles that allow eukaryotes to perform oxygenic photosynthesis. Oxygenic photosynthesis (hereafter simply photosynthesis) evolved in cyanobacteria around 2.4 billion years ago (bya), leading to the Great Oxidation Event—a rise of oxygen that profoundly transformed the Earth's atmosphere and shallow ocean[1,2]. Eukaryotes later acquired the capacity to photosynthesise with the establishment of plastids by endosymbiosis. Plastids originated from primary endosymbiosis between a cyanobacterium and a heterotrophic eukaryotic host, leading to primary plastids in the first photosynthetic eukaryotes. There are three main lineages with primary plastids: red algae, green algae (including land plants) and glaucophytes—altogether forming a large group known as Archaeplastida[3,4]. Subsequently, to the primary endosymbiosis, plastids spread to other eukaryote groups from green and red algae by eukaryote-to-eukaryote endosymbioses, i.e. the uptake of primary plastid-containing algae by eukaryotic hosts. These higher-order endosymbioses resulted in complex plastids surrounded by additional membranes, some even retaining the endosymbiont nucleus (the nucleomorph) and led to the diversification of many photosynthetic lineages of global ecological importance, especially those with red algal-derived plastids (e.g. diatoms, dinoflagellates and apicomplexan parasites)[5].

The evolution of complex red algal-derived plastids has been difficult to decipher, mainly because the phylogeny of host lineages does not straightforwardly track the phylogeny of plastids. From the plastid perspective, phylogenetic and cell biological evidence supports a common origin of all complex red plastids[6–11]. This is at the centre of the chromalveolate hypothesis[12], which proposed that the series of events needed to establish a plastid is better explained by a single secondary endosymbiosis in the common ancestor of alveolates, stramenopiles, cryptophytes and haptophytes: the four major groups known to harbour complex red plastids. From the host side, however, the phylogenetic relationships of these four groups have become increasingly difficult to reconcile with a single origin of

all complex red algal-derived plastids in a common ancestor. Indeed, over a decade of phylogenomic investigations have consistently shown that all red plastid-containing lineages are most closely related to a series of plastid-lacking lineages, often representing several paraphyletic taxa, which would require extensive plastid losses under the chromalveolate hypothesis (at least ten)[5]. This situation is further complicated by the fact that no cases of complete plastid loss have been demonstrated, except in a few parasitic taxa[13,14].

The current phylogeny of eukaryotes has given rise to a new framework for explaining the distribution of complex red plastids. This framework, unified under the rhodoplex hypothesis, invokes the process of serial endosymbiosis, specifically a single secondary endosymbiosis between a red alga and a eukaryotic host, followed by successive higher-order—tertiary, quaternary—endosymbioses spreading plastids to unrelated groups[15]. Several models compatible with the rhodoplex hypothesis have been proposed, differing in the specifics of the plastid donor and recipient lineages[16–19] (Fig. 1). However, these models of serial endosymbiosis remain highly speculative, in particular, because we do not know if they are chronologically possible—did the plastid donor and recipient lineages co-exist? Addressing this important issue requires a reliable timeframe for eukaryote evolution, which has been challenging to obtain owing to a combination of complicating factors, notably: (1) uncertain phylogenetic relationships among the major eukaryote lineages, (2) the lack of genome-scale data for the few microbial groups with a robust fossil record, and (3) a generally poor understanding of methodological choices on the dates estimated for early eukaryote evolution.

Recent molecular clock analyses placed the origin of primary plastids in an ancestor of Archaeplastida in the Paleoproterozoic Era, between 2.1–1.6 bya[20]. The origin of red algae has been estimated in the late Mesoproterozoic to early Neoproterozoic (1.3–0.9 bya)[20], after a relatively long lag following the emergence of Archaeplastida. However, an earlier appearance in the late

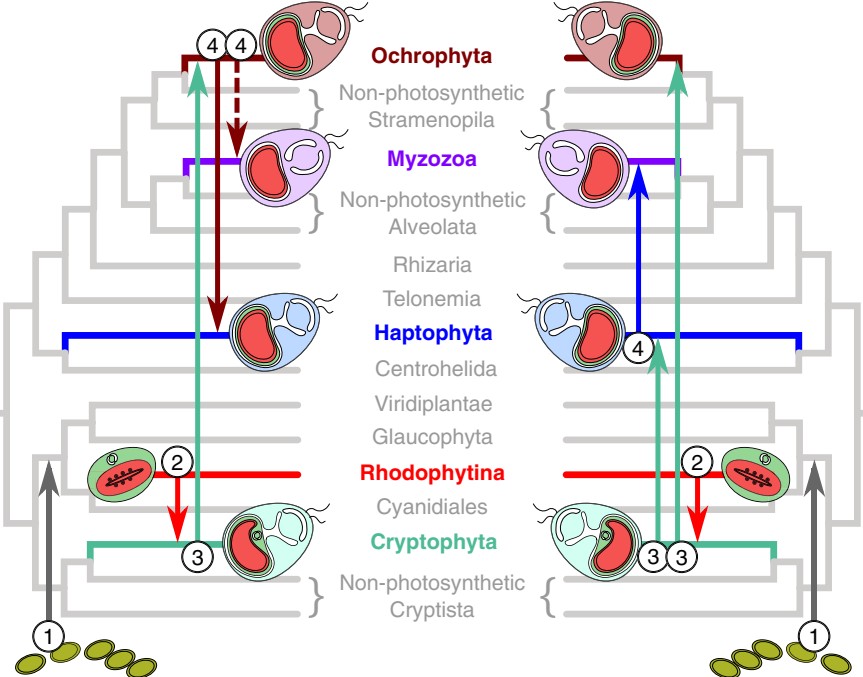

**Fig. 1 Serial plastid endosymbioses models as proposed by Stiller et al.[18] (left) and Bodył et al.[46] (right).** The tree topology shown here is based on the results obtained in our study. Further models have been suggested but are not compatible with this topology[15,117,118]. Numbers denote the level of endosymbiosis events. Note, Myzozoa were not included by Stiller et al.[18] and the dashed line indicates engulfment of an ochrophyte by the common ancestor of Myzozoa as suggested by Sevcikova et al.[19].

Paleoproterozoic has also been proposed based on molecular analyses[21,22]. The earliest widely accepted fossil for the crown-group Archaeplastida is the multicellular filamentous red alga *Bangiomorpha* deposited ~1.2 bya[23]. Recently, older filamentous fossils (*Rafatazmia* and *Ramathallus*) interpreted as crown-group red algae were recovered in the ~1.6 Ga old Vindhyan formation in central India[24]. This taxonomic interpretation pushed back the oldest commonly accepted red algal fossil record by 400 million years, and consequently the origin of red algae and Archaeplastida well into the Paleoproterozoic Era. Evidence for the diversification of red algal plastid-containing lineages comes much later in the fossil record, which is apparent in the well-documented Phanerozoic continuous microfossil records starting from about 300 million years ago (mya). This period marks the diversification of some of the most ecologically important algae in modern oceans such as diatoms (ochrophytes), dinoflagellates (myzozoans) and coccolithophorids (haptophytes). If the fossil record is taken at face value, there is, therefore, a gap of over one billion years between the first appearance of crown-group eukaryotes interpreted as red algae in the early Mesoproterozoic and the later rise to ecological prominence of red algal-derived plastid-containing lineages[25].

In this study, we combine phylogenomics and molecular clock analyses to investigate the chronology of the origin and spread of complex red plastids among distantly-related eukaryote lineages in order to test the general rhodoplex hypothesis[15]. We assemble a broad gene- and taxon-rich dataset (320 nuclear protein-coding genes, 733 taxa), incorporating 33 well-established fossil calibrations, to estimate the timing of early eukaryote diversification. We explore the effect of a range of Bayesian molecular clock implementations, relaxed clock models and prior calibration densities, as well as two alternative roots for the eukaryote tree. Our analyses show that the hypotheses of serial endosymbiosis are chronologically possible, as most red algal plastid acquisitions likely occurred in an overlapping timeframe during the Mesoproterozoic and Neoproterozoic Eras, setting the stage for the subsequent evolution of the most successful algae on Earth.

## Results

**The phylogeny of eukaryotes.** Molecular clock analyses rely on robust tree topologies. To obtain our reference topology, we derived two sub-datasets from the full dataset of 320 protein-coding genes and 733 taxa (Supplementary Fig. 1) to allow computationally intensive analyses: a 136-OTU dataset and a 63-OTU dataset (see 'Methods'). The 136-OTU dataset was used in maximum likelihood (ML) inference using the best fitting site-heterogeneous LG + C60 + G + F-PMSF model (hereafter simply *ML-c60*) based on a concatenated alignment, as well as with a supertree method consistent with the Multi Species Coalescent model (MSC) in a version of the alignment where the taxa within OTUs were retained individually. The concatenated 63-OTU dataset was analysed by Bayesian inference using the site heterogeneous CAT + GTR + G (hereafter simply *catgtrg*) and LG + C60 + G + F (*BI-c60*) models, as well as in posterior predictive analyses (PPA) to compare the fit of both models and after reducing compositional heterogeneity.

The tree based on the full dataset was in good overall agreement with the current consensus of the broad eukaryote phylogeny and classification[26], despite including some highly incomplete and fast-evolving taxa and being derived from the site-homogeneous LG + G + F model; site-homogeneous models do not capture site-specific amino acid preference and as a result, can cause systematic errors in phylogenetic estimation[27]. As expected from this model and such a heterogeneous taxon-sampling, the deeper nodes were generally unsupported and we

observed putative cases of long-branch attraction, for instance, the grouping of Metamonada, Microsporidia and Archamoebae (Supplementary Fig. 1). However, lateral gene transfers among these groups may also account for their grouping[28]. The more robust *ML-c60* tree derived from the 136-OTU dataset recovered many proposed supergroups with maximal bootstrap support (Supplementary Fig. 2), namely the TSAR assemblage, Haptista, Cryptista, Discoba, Amoebozoa and Obazoa[26,29,30]. Archaeplastida was also recovered monophyletic, albeit with lower bootstrap support (86%), but this supergroup previously lacked support in phylogenomic analyses (see 'Discussion'). The relationships among these supergroups were also consistent with published work, most notably the recurrent affinity between Cryptista and Archaeplastida (CA clade), the branching of Haptista with *Ancoracysta twista* in ML analyses and the placement of this group deep in the tree (here sister to the CA clade with 95% bootstrap support). The MSC analyses mostly recapitulated the same observations, although with the exceptions of TSAR and Archaeplastida due to the unresolved positions of telonemids, as well as red algae and Cryptista, respectively (Supplementary Fig. 3). Taken together, the MSC analyses either supported the results of the concatenated ML analysis, or were inconclusive rather than conflicting.

The *catgtrg* tree based on the 63-OTU dataset (Supplementary Fig. 4) received maximal posterior probabilities (PP) for all bifurcations; it is nearly fully consistent with the *ML-c60* analyses (Supplementary Fig. 2), albeit with one important difference for understanding plastid evolution: in the *catgtrg* tree, *A. twista* was inferred as sister to the group containing Haptista and TSAR. A Bayesian reanalysis of the 63-OTU dataset under the *BI-c60* model—the same model as in ML—recapitulated the *ML-c60* topology, suggesting that the position of *A. twista* was influenced by the evolutionary model rather than the use of Bayesian or ML inference as has been observed before[31] (Supplementary Fig. 5). The *catgtrg* topology was also not rejected by ML in an Approximately Unbiased test (p-AU = 0.182), providing additional support for this tree. Furthermore, we used posterior predictive tests to determine which model better minimises inadequacy in describing compositional heterogeneity and found that *catgtrg* is superior to *BI-c60*, although neither model fully described the data (Supplementary Table 1). Thus, to help reducing compositional heterogeneity, we performed site-stripping of the compositionally most biased sites. The 25% and 50% most compositionally heterogeneous sites were stripped from the 63-OTU alignment, and trees were reconstructed with *catgtrg* (Supplementary Fig. 6). Both analyses fully confirmed the *catgtrg* tree recovered from the full-length alignment, with only a minor exception in the position of the apusozoan *Nutomonas*, which moved sister to Discoba in the shortest alignment (Supplementary Fig. 6b).

Finally, we evaluated whether our selected phylogenetic markers displayed signal resulting from potential endosymbiotic gene transfers (EGTs) between the endosymbiont and host genomes during plastid establishment[9]. Relationships among algae might be affected by EGT, which, if undetected, would distort the species tree and compromise our efforts to test hypotheses of red plastid spread by reference to the host phylogeny. For example, the inferred sister relationship between Cryptista and Archaeplastida could be an artefact due to the replacement of host cryptophyte genes by homologues from the red algal endosymbiont. We systematically evaluated bootstrap support for sister-group relationships between each red plastid-containing lineage and all other eukaryotic taxonomic groups for each of the 320 marker genes independently ('Methods'). This analysis provided no positive evidence for horizontal acquisition of any marker genes during evolution, as we did not detect

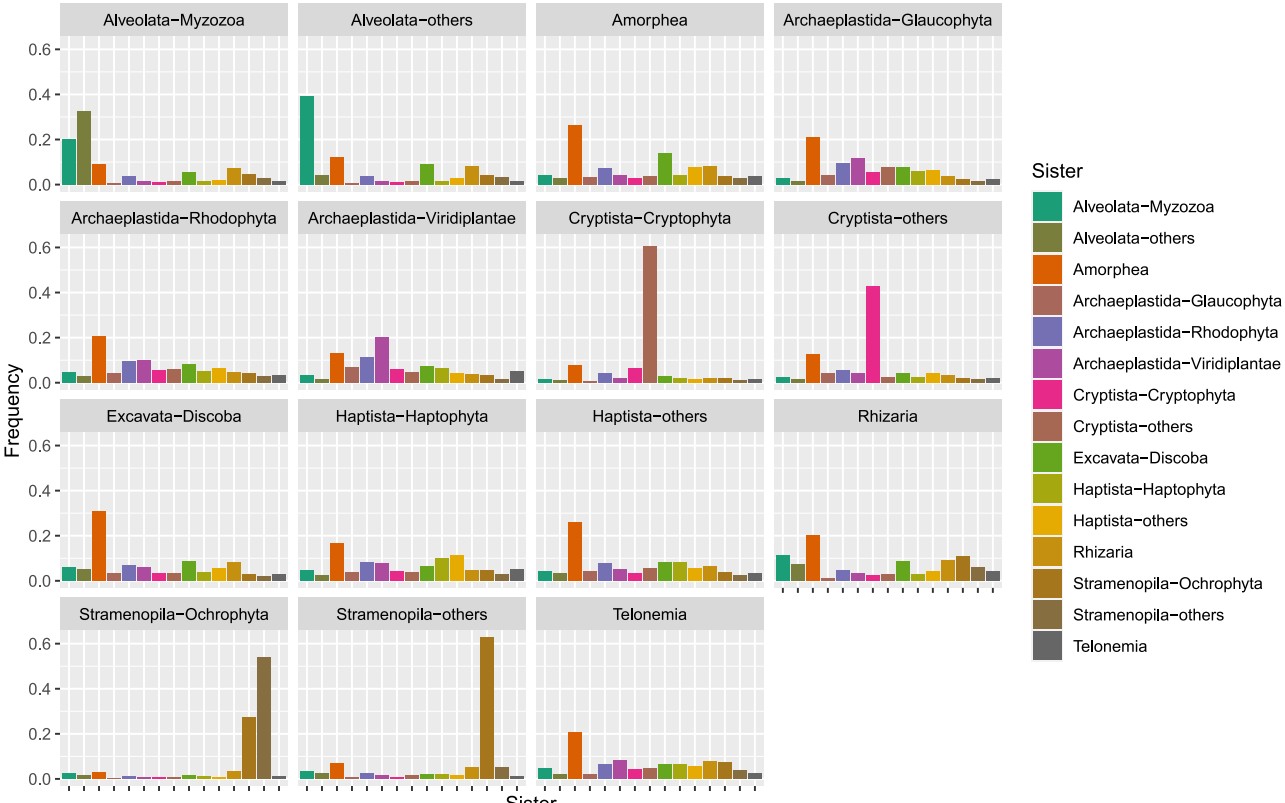

**Fig. 2 Test for endosymbiotic gene transfers in red plastid-containing lineages based on the analysis of 320 single-gene ML trees.** For each clade in the ML tree to which a single taxonomic label could be assigned, relative frequencies with which all other clades in the tree were recovered as the closest sister group are given (for details, see 'Methods').

dominant non-vertical signals for any group. That is, bootstrap support from single genes was either equivocal for deep relationships, or favoured the branching of the lineage that would be expected based on prior knowledge of species relationships (Fig. 2). Note that, since this analysis relies on gene tree conflict to identify EGTs, a lineage in which most or all marker genes have been replaced by EGT might not be detected.

Taken together, these results favoured the relationships described under the *catgtrg* model, which was less affected by obvious biases. Therefore, the backbone topology for the divergence time estimation discussed below was inferred under a set of constraints defined by the best *catgtrg* topology but using the 136-OTU dataset, which contains a broader taxon sampling allowing to more precisely place fossil calibrations.

**Timescale for eukaryote evolution.** Our molecular clock analyses revealed congruent dates inferred under comparable analytical conditions (i.e. clock models, prior calibration densities and root positions; see below and 'Methods') and by all tested implementations (MCMCTree, PhyloBayes, BEAST). The fossils used as calibrations were chosen to span a wide diversity of lineages and ages (Table 1), and we followed a conservative approach when interpreting the fossil record by choosing only fossils widely accepted by the palaeontological community. We also included one generally uncontested biomarker to constrain the emergence of extant Metazoa in order to expand the otherwise sparse Proterozoic fossil record (24-isopropyl cholestane)[32]. Our analyses placed the root of the eukaryote tree well into the Paleoproterozoic Era (Fig. 3). This Era also saw the origin of primary plastids in the common ancestor of Archaeplastida, which likely took place between 2137 and 1807 mya. Crown group red algae were inferred in the Paleoproterozoic between 1984 and 1732 mya.

These age ranges are provided as a conservative approach encompassing all performed analyses (with the exception of those using t-cauchy distributions with long tails, see below).

From red algae, plastids then spread to distantly-related groups of eukaryotes by at least one secondary endosymbiosis. Plastid phylogenies have consistently shown that this red algal donor lineage belonged to a stem lineage of Rhodophytina[33], i.e. it lived after the split of Cyanidiophyceae and the rest of red algae (Fig. 3); we inferred a time range for this donor lineage to be between 1675 and 1281 mya. The origination period for the lineages currently harbouring red algal-derived plastids was inferred as follows: cryptophytes between 1658 and 440 mya; ochrophytes between 1298 and 622 mya; haptophytes between 1943 and 579 mya; myzozoans between 1520 and 696 mya (dates refer to the 95% HPD intervals in Fig. 3). Thus, the stems of all extant lineages containing red algal plastids—along which these plastids were acquired—overlap chronologically. This overlap was consistent across all performed analyses and defines the time windows during which endosymbiotic transfers, as proposed by the rhodoplex framework, could reconcile plastid and nuclear phylogenies (Figs. 3 and 4; Supplementary Figs. 7, 8 and Supplementary Data 1). Note that PhyloBayes analyses with the autocorrelated clock model displayed the shortest confidence intervals, while the MCMCTree analyses with the same clock model recovered generally younger (i.e. closer to present) ages for the origin of ochrophytes and myzozoans (Fig. 4), but these differences did not alter the observation of time overlap for the putative plastid acquisitions.

To better understand the effect of methodological choices on divergence times, we performed a battery of sensitivity analyses with MCMCTree that tested different combinations of clock models, prior calibration densities, the position of the eukaryote root, as well

**Table 1 Calibrations used for dating the eukaryote tree of life.**

| Clade | Age constraint (Ma) | Type | Eon | Refs. |
|---|---|---|---|---|
| Rhodophyta[a] | 1600 | Min | Proterozoic | [24] |
| Bangiophyceae/Florideophyceae | 1600 | Max | Proterozoic | [b] |
| Bangiophyceae/Florideophyceae | 1047 | Min | Proterozoic | [23,90] |
| Metazoa | 833 | Max | Proterozoic | [91] |
| Ciliophora[a] | 740 | Min | Proterozoic | [92] |
| Euglyphidae[a] | 736 | Min | Proterozoic | [93,94] |
| Evosea/Tubulinea[a] | 736 | Min | Proterozoic | [94,95] |
| Tubulinea | 736 | Max | Proterozoic | [b] |
| Chlorophyta[a] | 700 | Min | Proterozoic | [96] |
| Eumetazoa | 636 | Max | Proterozoic | [91] |
| Deuterostomia | 636 | Max | Proterozoic | [91] |
| Chordata | 636 | Max | Proterozoic | [91] |
| Bilateria | 636 | Max | Proterozoic | [91] |
| Arthropoda | 636 | Max | Proterozoic | [91] |
| Metazoa | 635 | Min | Proterozoic | [32] |
| Eumetazoa | 550 | Min | Proterozoic | [91] |
| Bilateria | 550 | Min | Proterozoic | [91] |
| Mollusca | 549 | Max | Proterozoic | [91] |
| Foraminifera[a] | 542 | Min | Proterozoic | [97,98] |
| Bacillariophyta | 541 | Max | Proterozoic | [99] |
| Embryophyta | 540 | Max | Phanerozoic | [20,100] |
| Mollusca | 532 | Min | Phanerozoic | [91] |
| Deuterostomia | 515 | Min | Phanerozoic | [91] |
| Chordata | 514 | Min | Phanerozoic | [91] |
| Arthropoda | 514 | Min | Phanerozoic | [91] |
| Embryophyta | 470 | Min | Phanerozoic | [101] |
| Angiosperms/Gymnosperms | 470 | Max | Phanerozoic | [b] |
| Euglenales/Eutreptiales[a] | 450 | Min | Phanerozoic | [102] |
| Chytridiomycota[a] | 410 | Min | Phanerozoic | [103,104] |
| Tubulinea | 405 | Min | Phanerozoic | [105] |
| Ascomycota[a] | 400 | Min | Phanerozoic | [106] |
| Angiosperms/Gymnosperms | 385 | Min | Phanerozoic | [107] |
| Angiosperms | 385 | Max | Phanerozoic | [b] |
| Basidiomycota[a] | 360 | Min | Phanerozoic | [108] |
| Amniota | 333 | Max | Phanerozoic | [91] |
| Amniota | 318 | Min | Phanerozoic | [91] |
| Core dinoflagellates (excl. Noctilucales) | 300 | Max | Phanerozoic | [49] |
| Coccolithales/Isochrysidales | 260 | Max | Phanerozoic | [49] |
| Core dinoflagellates (excl. Noctilucales) | 235 | Min | Phanerozoic | [109] |
| Peridiniales | 235 | Max | Phanerozoic | [b] |
| Gonyaulacales | 235 | Max | Phanerozoic | [b] |
| Coccolithales/Isochrysidales | 225 | Min | Phanerozoic | [110] |
| Calcidiscaceae/Coccolithaceae | 225 | Max | Phanerozoic | [b] |
| Peridiniales | 210 | Min | Phanerozoic | [109] |
| Gonyaulacales | 200 | Min | Phanerozoic | [109] |
| Bacillariophyta | 190 | Min | Phanerozoic | [99] |
| Pennales | 190 | Max | Phanerozoic | [b] |
| Euarchontoglires | 165 | Max | Phanerozoic | [91] |
| Angiosperms | 130 | Min | Phanerozoic | [111] |
| Eudicotyledons (Tricoplates) | 130 | Max | Phanerozoic | [b] |
| Eudicotyledons (Tricoplates) | 124 | Min | Phanerozoic | [112,113] |

**Table 1 (continued)**

| Clade | Age constraint (Ma) | Type | Eon | Refs. |
|---|---|---|---|---|
| Aves sensu stricto | 87 | Max | Phanerozoic | [91] |
| Pennales | 75 | Min | Phanerozoic | [99] |
| Aves sensu stricto | 66 | Min | Phanerozoic | [91] |
| Calcidiscaceae/Coccolithaceae | 65 | Min | Phanerozoic | [114,115] |
| Euarchontoglires | 61 | Min | Phanerozoic | [91] |

[a]Max = 1900 Ma was used; see for example Knoll[116] and Eme et al.[48].
[b]Max based on Min age of ancestor.

as the effect of removing the oldest calibration on red algae (36 sensitivity analyses in total; Supplementary Data 1 and 'Methods'). These analyses indicated that the prior calibration distributions had the strongest effect on the inferred divergence times, followed by the clock model, the root position and the removal of the red algal calibration. Prior calibrations model the uncertainty of fossil ages and their proximity to the cladogenetic events being calibrated, and thus different distributions can be understood as more literal (skew-normal), loose (t-cauchy) or conservative (uniform) interpretations of the fossil record[34]. As expected from the prior distributions, we observed younger overall ages with skew-normal calibrations (median of 806 Ma) compared to uniform (median of 823 Ma) or t-cauchy distribution with short or long tails (median of 1082 and 1551 Ma, respectively; Supplementary Data 1). The excessively old ages and wide 95% HPD intervals (median of 500 vs. 322 to 397 Ma) inferred with long-tailed t-cauchy distribution were considered biologically implausible, and thus disregarded in the following. The clock model had a modest impact on the posterior dates, with the autocorrelated clock model generally producing slightly younger ages and narrower intervals (median ages 1004 Ma; 95% HPD median widths 356 Ma) than the uncorrelated clock (median ages 1025 Ma and 95% HPD median widths 432 Ma). The younger ages inferred under the autocorrelated clock model were most apparent when uniform calibrations were applied (median ages of 765 vs. 926 Ma). CorrTest[35] indicated that branch lengths are most likely correlated (CorrScore = 0.99808, $p < 0.001$), suggesting that autocorrelated models might better model our dataset. The use of two alternative roots, either on Amorphea or on Discoba, had a small effect on the posterior ages. Only marginal differences were observed on the overall node median times (1015 vs. 1008 Ma for the Amorphea and Discoba roots, respectively) and median interval widths (393 vs. 391 Ma). The only exceptions were the basal relationships within Discoba, which were noticeably older when rooting the tree on this group (Fig. 3; Supplementary Fig. 7). Finally, the removal of the oldest calibration for the crown-group of red algae, set at 1600–1900 Ma[24], shifted most (82%) node ages towards present by an average of 127 Ma under the autocorrelated clock model, while the age differences were unappreciable under the uncorrelated clock model (mean of 6 Ma across all nodes). Importantly, however, the 95% HPD intervals remained overlapping between the red algal plastid donor lineage and the origination periods of all lineages with red-complex plastids, suggesting that our inferences regarding the rhodoplex hypothesis are robust to varying interpretation of this ancient Proterozoic fossil (Supplementary Fig. 9).

Further sensitivity analyses were performed with PhyloBayes to confirm the effects of the clock model choice (Supplementary Data 1). We also tested the effect of the substitution model by comparing LG + G with the *catgtrg* mixture model. We observed a slightly higher impact of the evolutionary model than the clock model (median differences of 578 vs. 528 Ma, respectively). In this

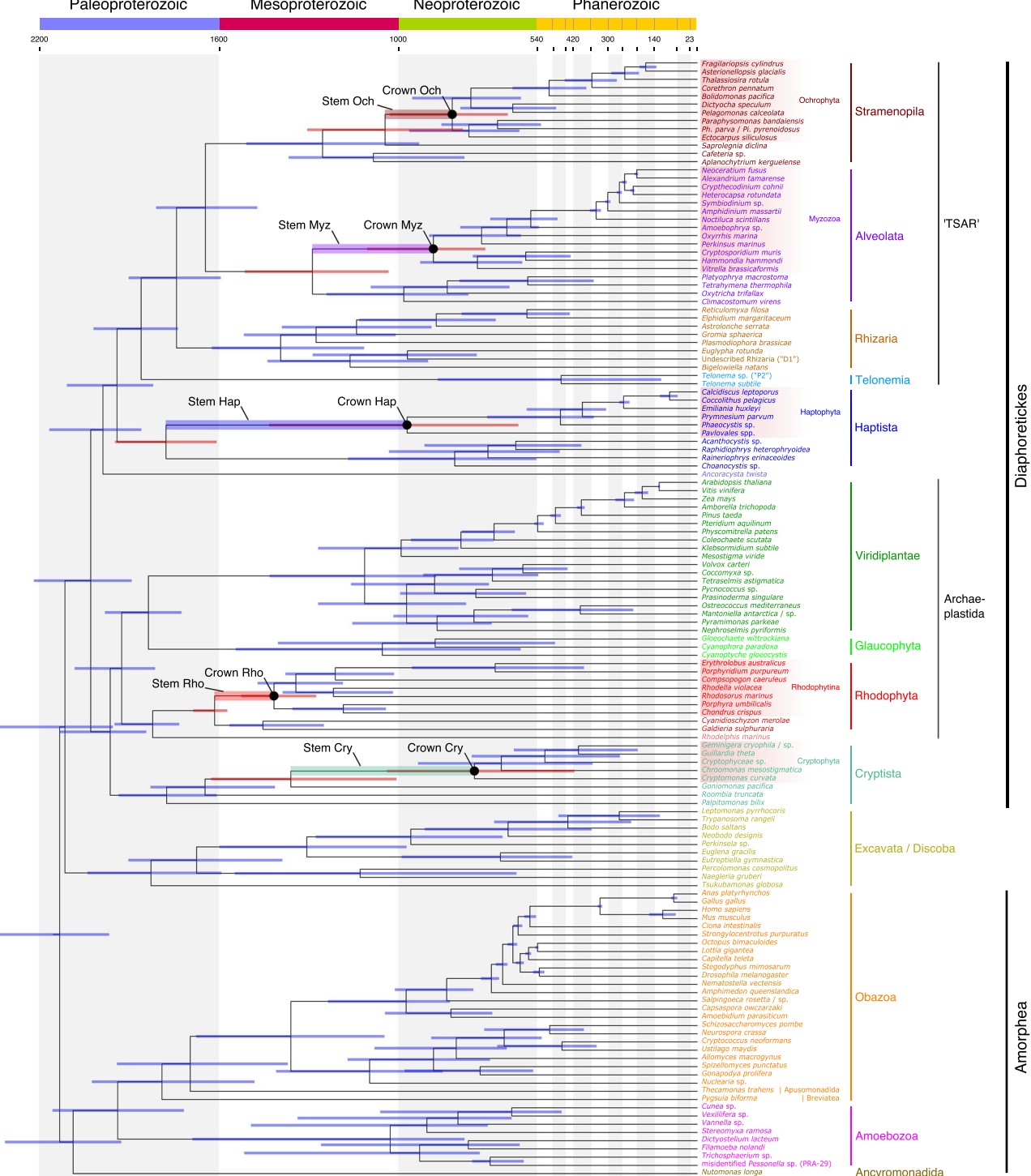

**Fig. 3 Time-calibrated phylogeny of extant eukaryotes.** Divergence times were inferred with MCMCTree under an autocorrelated relaxed clock model and 33 fossil calibration points as soft-bound uniform priors (Table 1). The tree topology was reconstructed using IQ-TREE under the LG + C60 + G + F model and a constrained tree search following the OTU-reduced Bayesian CAT + GTR + G topology (Supplementary Fig. 4). Approximate likelihood calculations on the 320 gene concatenation under LG + G and a birth–death tree prior were used. Bars at nodes are 95% HPD. Bars corresponding to the first and last common ancestors of extant red plastid-donating and -containing lineages are highlighted in red and their stems are shaded as indicated. Crowns denote the common ancestors of the extant members of these groups. An absolute time scale in Ma and a geological time scale are shown. The tree depicted here was rooted on Amorphea. An equivalent time-calibrated tree rooted on Excavata is shown in Supplementary Fig. 7. Cry Cryptophyta, Hap Haptophyta, Myz Myzozoa, Och Ochrophyta, Rho Rhodophytina.

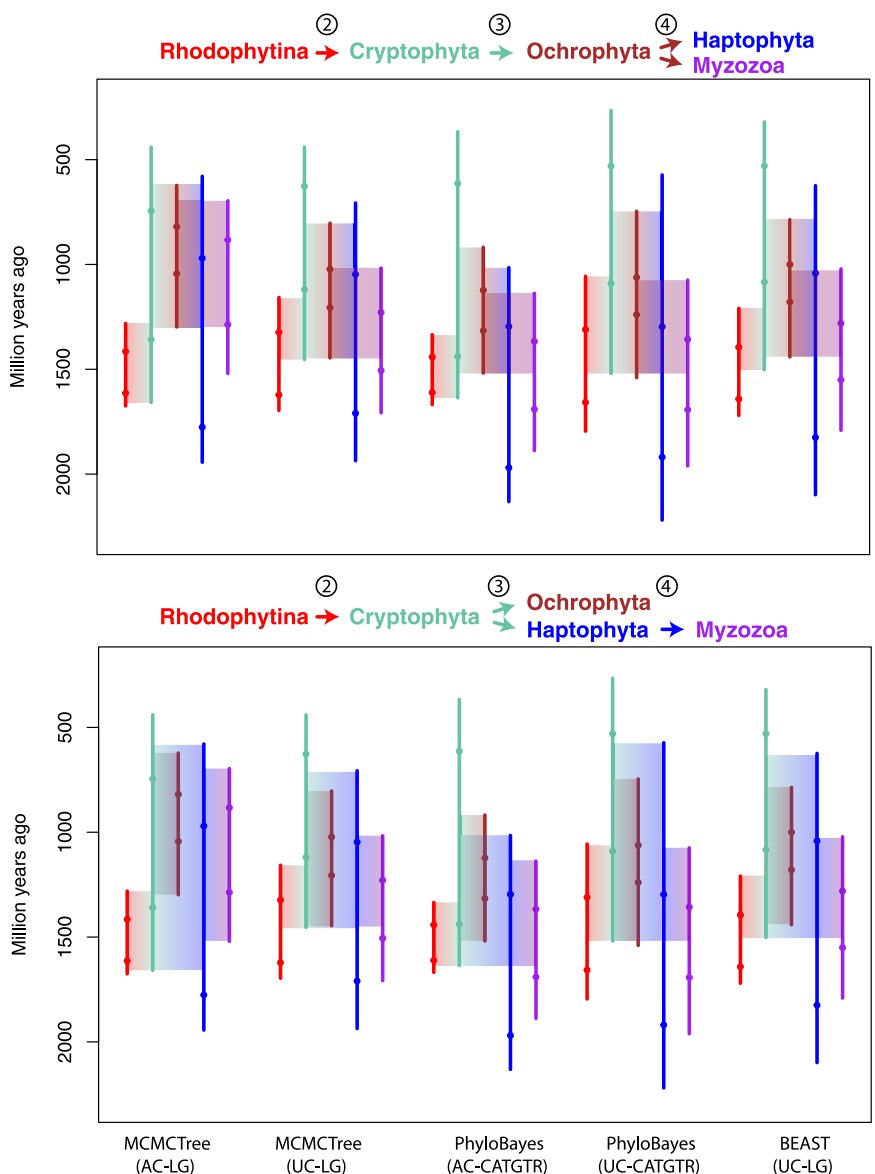

**Fig. 4 Summary of inferred timeframes for the spread of complex red plastids using different software and models.** Vertical lines correspond to the lower and upper 95% HPD intervals from the nodes defining the branch of interest and dots indicate their posterior mean divergences. Faded boxes represent the temporal windows for the secondary endosymbioses, constrained by the 95% HPDs and tree topology under the two proposed symbiotic scenarios. Numbers at arrows denote the level of endosymbiosis events (compare with Fig. 1). AC autocorrelated clock model, UC uncorrelated clock model.

case, *catgtrg* led to older ages and wider 95% HPD intervals than the LG + G model (median ages of 1123 vs. 1016 Ma and median 95% HPD widths of 388 vs. 354 Ma, respectively, under the auto-correlated model). The autocorrelated clock model tended to infer older ages and wider intervals than the uncorrelated model (e.g. median ages of 1123 vs. 978 Ma and median 95% HPD widths of 388 vs. 354 Ma under the *catgtrg* model). Finally, we tested a third commonly used Bayesian implementation (BEAST) using the LG + G substitution model, with the Amorphea root and uniform prior calibrations. The obtained median posterior ages and 95% HPD intervals were in overall agreement with those obtained by MCMCTree and PhyloBayes (Supplementary Data 1), corroborating the previously inferred dates.

## Discussion

**The eukaryote tree of life and the rhodplex hypothesis.** In the last 15 years, the tree of eukaryotes has been extensively remodelled based on phylogenomics. We assembled a dataset containing a dense eukaryotic-wide taxon sampling with 733 taxa and analysed subsets of it with a variety of mixture models to resolve several important uncertainties that are key for understanding plastid evolution. Notably, we recovered the monophyly of Archaeplastida, which has previously been strongly supported by plastid evidence e.g.[11,36], but not by host (nuclear) phylogenetic markers e.g.[37–40]. This topology is consistent with a recent exhaustive phylogenomic study of nuclear markers[4], as well as with the long-held view of a single point of entry of photosynthesis in eukaryotes from cyanobacteria through the establishment of primary plastids[41] (with the exception of the chromatophores in *Paulinella*[42]). The supergroup Cryptista, which includes the red algal plastid-containing cryptophytes, was placed as sister to Archaeplastida. The association of Cryptista and Archaeplastida has been suggested before, often even disrupting the monophyly of Archaeplastida[39,40], but our analyses robustly recovered the sister relationship of these two major

eukaryote clades. Another contentious placement has concerned the supergroup Haptista[29,39,40,43], which includes the red algal plastid-containing haptophytes, as well as its possible relationship to the orphan lineage *Ancoracysta twista*[31]. The better-fitting *catgtrg* model favoured the position of Haptista as sister to TSAR and placed *A. twista* deeper in the tree, not directly related to Haptista. We observed that the model (*catgtrg* vs. *lgc60*) rather than the phylogenetic method or software is determinant in placing *A. twista* relative to Haptista—*lgc60* always placed these two lineages together, both in ML and Bayesian, but *catgtrg* never did—and that variation in the taxon-sampling did not drastically modify this association.

Given this well-resolved tree of eukaryotes, we then mapped the position of the groups with red algal-derived plastids (Fig. 3). The rhodoplex hypothesis explains the distribution of these plastids by a series of endosymbioses, positing that plastids were separately acquired in the stem lineages of cryptophytes, ochrophytes, haptophytes and myzozoans[15]. While the rhodoplex hypothesis remains speculative, it provides the benefit of reconciling the accumulating discrepancies between plastid and host phylogenies that existed under the previously prevalent chromalveolate hypothesis[12]. In the chromalveolate hypothesis, a red plastid was acquired by secondary endosymbiosis in the common ancestor of all red plastid-bearing lineages, and multiple subsequent losses were invoked to explain the patchy distribution of red plastids across the eukaryote tree. Our phylogeny is consistent with other recent analyses in rendering this scenario impossible[11,19,29,39,40]: since Cryptista branches as sister to Archaeplastida, and the other red plastid lineages branch elsewhere in the tree, the common ancestor of red plastid-bearing lineages corresponds to one of the earliest nodes in the tree (Fig. 3), which was ancestral to the cell that acquired primary plastids at the origin of Archaeplastida. As it has been pointed out before[39], this scenario would require a red alga to travel backwards in time to be engulfed by one of its distant ancestors. In contrast, our analyses indicate that the inferred red algal donor lineage (stem Rhodophytina) was contemporaneous with stem cryptophytes, haptophytes and myzozoans, but not with stem ochrophytes, which in all likelihood had not yet diverged from other stramenopiles at the time (although their 95% HPDs marginally overlap). Moreover, all red algal plastid-containing lineages are most closely related to lineages without plastids (Fig. 3), and for which conclusive evidence for a past photosynthetic history does not exist. This situation would require many plastid losses if red plastids had been established early and vertically transmitted, indicating that the chromalveolate hypothesis no longer provides a compelling explanation for the distribution of red plastids in extant eukaryotes[5].

The rhodoplex hypothesis also allows reconciling non-phylogenetic plastid data with that of their hosts. The mechanism of protein import into complex red plastids involves a unique translocation machinery known as SELMA, which is derived from the ER-associated protein degradation (ERAD) system of the red algal endosymbiont[44]. SELMA has been interpreted as a robust character supporting a single plastid origin in a chromalveolate ancestor[36], but the presence of this machinery in all algae with red plastids (with the exception of myzozoans) is also compatible with serial acquisitions[5]. The molecular components of SELMA are nucleus- or nucleomorph-encoded in all investigated organisms, which implies that these genes would have been repeatedly transferred to the host nucleus during each rhodoplex endosymbiosis. Although this may appear less likely than a single origin of SELMA followed by vertical inheritance, it is worth noting that a similar mechanism of independent nuclear relocalisations of homologues of the TIC/TOC protein import machinery took place in the green lineages chlorarachniophytes and euglenids, as

well as in the red algal plastid-containing lineages[9]. Thus, the possibility exists that SELMA has also been successively reestablished during the process of serial endosymbiosis. Another plastid character that is in apparent contradiction with our host-derived topology is the horizontally transferred bacterial *rpl36* gene into the plastid genomes of haptophytes and cryptophytes[45]. Here again, the rhodoplex hypothesis is compatible with the existence of a specific link between the haptophyte and cryptophyte plastids but not between the hosts. In fact, this is an explicit possibility in the model of Bodyl et al.[46,47], which proposed that plastids were transferred twice from cryptophytes: once to ochrophytes before the *rpl36* lateral gene transfer and then to haptophytes after the *rpl36* replacement by a bacterial homologue.

**Timing the early evolution of eukaryotes**. We present a detailed molecular clock analysis providing a timeframe for eukaryote evolution. Our estimates inferred an age for the Last Eukaryote Common Ancestor (LECA) between 2386 and 1958 mya, which is generally older than in other molecular clock analyses based on phylogenomic data[48–50]. However, an early Paleoproterozoic origin of extant eukaryotes fits with the oldest definitive crown-eukaryote fossils, the putative red algae *Rafatazmia chitrakootensis* and *Ramathallus lobatus* from 1600 mya[24], which imply that eukaryotes must have originated before this time. It also fits with the recent discoveries of multicellular eukaryotes in different groups of Proterozoic fossils indicating that eukaryotes were already complex in deep times. For example, the chlorophyte fossil *Proterocladus antiquus* in ~1000 Ma old rocks, taken as evidence for a much earlier appearance of multicellularity in this group of green algae[51], is in line with our results (Fig. 3). Similarly, multicellular organic-walled microfossils with affinity to fungi were recently reported in the 1–0.9 Ga old Grassy Bay Formation[52], which pushes back the emergence of fungi by 500 Ma compared to the previous studies[53]. In our analyses, fungi were estimated to originate even before (1759 to 1078 Ma), consistently with the presence of multicellular organisms around 1 bya. More generally, an early Paleoproterozoic origin of eukaryotes would also be in line with records of aggregative multicellularity appearing more than 2 bya, although the eukaryote affiliation of these fossils is debated[54,55].

Our estimated dates placed the common ancestor of Archaeplastida also in the Paleoproterozoic Era, suggesting that eukaryotes acquired primary plastids, and thus photosynthesis, early on. This early origin of plastids is consistent with the putative photosynthetic crown-Archaeplastida acritarch *Tappania*, which offers circumstantial evidence that eukaryotes exploited photosynthesis from the earliest period of eukaryote evolution[56]. It is also in agreement with a recent molecular clock analysis of early photosynthetic eukaryotes that placed the origin of Archaeplastida at ~1900 mya[20]. Furthermore, total group red algae are generally thought to contain some of the best minimum constraints for the existence of fully photosynthetic eukaryotes, most convincingly the late Mesoproterozoic *Bangiomorpha* resembling extant large bangiophytes[23], but also the earlier *R. chitrakootensis* and *R. lobatus*, probably also multicellular from 1600 Ma[24]. Thus, both the fossil record and molecular clock inferences support a Paleoproterozoic origin of primary plastids, and an early Mesoproterozoic origin of red algae.

For any endosymbiotic relationship to be established, the endosymbiont and the host must live at the same time and in the same place in order to interact. While the time intervals for plastid acquisition are sometimes relatively wide in our analyses (Fig. 3), they can be further constrained by overlaying the direction of plastid transfers proposed in the different models

**Table 2 Inferred time intervals for the acquisition of red algal-derived plastids by serial endosymbioses.**

| | MCMCTREE | | | | | | PHYLOBAYES | | | | | | BEAST | | |
| | AUTOCORRELATED | | | UNCORRELATED | | | AUTOCORRELATED | | | UNCORRELATED | | | UNCORRELATED | | |
| | max | min | width | max | min | width | max | min | width | max | min | width | max | min | width |
| Stiller et al. | | | | | | | | | | | | | | | |
| Rho => Cry | 1658 | 1281 | 377 | 1453 | 1158 | 296 | 1636 | 1335 | 301 | 1517 | 1057 | 461 | 1502 | 1210 | 292 |
| Cry => Och | 1298 | 623 | 675 | 1446 | 804 | 642 | 1518 | 918 | 600 | 1517 | 747 | 771 | 1441 | 786 | 654 |
| Och => Hap | 1298 | 623 | 675 | 1446 | 804 | 642 | 1518 | 1051 | 467 | 1517 | 747 | 771 | 1441 | 786 | 654 |
| Och => Myz | 1298 | 696 | 602 | 1446 | 1017 | 428 | 1518 | 1138 | 380 | 1517 | 1075 | 443 | 1441 | 1022 | 419 |
| Bodyl et al. | | | | | | | | | | | | | | | |
| Rho => Cry | 1658 | 1281 | 377 | 1453 | 1158 | 296 | 1636 | 1335 | 301 | 1517 | 1057 | 461 | 1502 | 1210 | 292 |
| Cry => Och | 1298 | 623 | 675 | 1446 | 804 | 642 | 1518 | 918 | 600 | 1517 | 747 | 771 | 1441 | 786 | 654 |
| Cry => Hap | 1658 | 579 | 1079 | 1453 | 707 | 747 | 1636 | 1051 | 585 | 1517 | 573 | 944 | 1502 | 624 | 878 |
| Hap => Myz | 1520 | 696 | 824 | 1453 | 1017 | 436 | 1636 | 1138 | 498 | 1517 | 1075 | 443 | 1502 | 1022 | 480 |

Time intervals have been calculated from overlapping 95% HPD intervals, constrained by the order of symbioses in the models of Stiller et al.[18] or Bodyl et al.[46]. Numbers are in million years. Cry Cryptophyta, Hap Haptophyta, Myz Myzozoa, Och Ochrophyta, Rho Rhodophytina.

congruent with the rhodoplex hypothesis (Fig. 1). Of these models, only those by Stiller et al.[18] and Bodyl et al.[46] are compatible with our tree (Fig. 3), because the other models assumed the monophyly of haptophytes with cryptophytes and the acquisition of plastids in an ancestor of that group. In both compatible models, the secondary engulfment of a red alga by a stem cryptophyte can be constrained by the minimum age of the plastid donor lineage, i.e. the age of the last Rhodophytina common ancestor (Fig. 3). This would indicate a rather early plastid acquisition by cryptophytes and a relatively long time span before the diversification of extant cryptophytes. In fact, if a red plastid-bearing cryptophyte was subsequently acquired by tertiary endosymbiosis by a stem ochrophyte, as both models proposed, this acquisition likely took place at latest 819 mya (95% HPD: 633–1017), thus providing a maximum constraint for the age of crown cryptophytes. The model of Bodyl et al.[46] also proposed that cryptophytes were acquired by an ancestor of haptophytes in a parallel tertiary endosymbiosis, thus resulting in further constraints on the cryptophyte stem lineage. However, phylogenetic analyses of plastid-targeted proteins have instead supported an ochrophyte origin for the haptophyte plastid[18]. Following the same logic, the most likely time ranges of plastid acquisitions in all four red algal plastid-containing lineages and for the models and data compatible with our tree are presented in Table 2. Strikingly, this approach to chronologically constraining red plastid endosymbioses revealed that the serial transfers all took place in a relatively short time window between 650 and 1079 million years from the initial secondary endosymbiosis in a stem cryptophyte to the establishment of plastids in the ancestors of all modern-day red algal plastid-containing lineages (narrowest and widest 95% HPD widths for the totality of the four endosymbioses; Table 2).

**The origin and rise of algae.** The fossil and biomarker records document the increasing abundance and rise to ecological prominence of primary endosymbiotic algae in the oceans, including red algae, ~659–645 mya[57]. Red plastid-bearing algae, notably diatoms, coccolithophores and dinoflagellates, started to expand later after the Permo-Triassic transition ~250 mya and have since remained major primary producers in the oceans[58]. By contrast, our molecular clock analyses indicate that the events underlying the evolutionary origin of these algae, including the primary plastid endosymbiosis, the origin of red algae and the subsequent spread of red plastids across the eukaryote tree, all pre-dated the ecological expansion of these groups by at least ~0.5–1 Ga. This long inferred period could be due, at least in part, to processes difficult to models, such as an early burst of evolutionary changes during endosymbiotic integration[59], or the general mosaic nature of algal genomes[9]. If real, however, this time lag suggests that the selective pressures underpinning the establishment of primary and complex plastids in the Palaeo- and Mesoproterozoic Eras are distinct from those that drove their expansion ~1 Ga later.

The Proterozoic ocean was poor in essential inorganic nutrients, such as phosphate and nitrogen, which may have limited the expansion of larger eukaryotic algae into the open marine realm[58]. Yet, these oligotrophic environments may have been favourable to the origin and early evolution of plastid-containing lineages. Predatory behaviours have been demonstrated in green algae and non-photosynthetic direct relatives of red algae, indicating that phagotrophy persisted alongside phototrophy for long evolutionary times in Archaeplastida and that mixotrophy was a key intermediate stage in the early evolution of plastids[60,61]. More generally, mixotrophy is increasingly recognised as the default lifestyle for many, perhaps most single-cell algae with complex plastids and is clearly advantageous

over both autotrophy or heterotrophy in communities limited by nutrients[62,63]. Thus, oligotrophic Proterozoic waters could have selected for the increased fitness that mixotrophy provided. The rise to ecological dominance of algae with red plastids during the Mesozoic might have been driven by profound environmental changes, such as the increase in coastlines associated with the breakup of the supercontinent Pangea, providing newly flooded continental margins with high-nutrient habitats[64]. Importantly, these environmental changes would have happened after ~1 Ga of evolution since the origin of red algal-derived plastids, providing ample opportunities for the evolution of a genetic toolkit that would prove beneficial with the rise of more favourable habitats. One such example of beneficial genetic innovation is cell protection by a variety of armour plating, which is a convergent feature of many ecologically successful algae with complex red plastids. These armours protect the phytoplankton from grazing and thus represent an additional condition that may have favoured the late Mesozoic expansion to ecological dominance of some groups with red algal-derived plastids[64].

In conclusion, algae powered by red algal-derived plastids are among the most evolutionary and ecologically successful eukaryotes on Earth. Yet, we still lack a comprehensive understanding of how, and how many times, red plastids were established. In recent years, hypotheses of serial endosymbiosis have flourished to explain how disparate groups of eukaryotes obtained their red plastids. In the present study, we used molecular clocks applied to a broad phylogenomic dataset to test whether the serial endosymbiosis hypotheses are chronologically possible. Our results indicate that all putative plastid donor and recipient lineages most likely overlapped during Earth history, thus in principle allowing plastids to be passed between distantly related hosts. Furthermore, we showed that the timeframe from the initial secondary endosymbiosis with a red alga to the establishment of all complex red plastids was relatively short, likely spanning between 650 to 1079 million years mainly during the Mesoproterozoic Era. This relatively short timeframe represents a novel insight into the diversification of photosynthetic eukaryotes during the Mesoproterozoic and the origin of the most ecologically important modern-day algae. More generally, specific serial endosymbiosis hypotheses, if validated, will provide useful relative constraints for better understanding the overall timescale of eukaryote diversification in future paleobiological studies.

## Methods

**Phylogenomic dataset construction**. Throughout this study, amino acid sequences were used for phylogenomic analyses. Two publicly available datasets were used as starting points: 263 protein-coding genes, 234 taxa dataset[29,39] and 351 protein-coding genes, 64 taxa dataset[65]. Non-overlapping genes between these two datasets (134 genes) were identified by BLAST[66] analyses, allowing the merging of both datasets to bring the total number of initial genes to 397. We expanded the sampling of species with publicly available genomic/transcriptomic data to obtain a comprehensive eukaryote-wide dataset, with particular attention to taxa most relevant to this study (sources: ensemblgenomes.org, imicrobe.us/#/projects/104, ncbi.nlm.nih.gov, onekp.com, and few publications that did not provide a link to a public sequence database; now all available in EukProt[67]). The procedure to add taxa was as follows: (1) For each taxon, protein sequences were clustered with CD-HIT[68] using an identity threshold of 85%, (2) Homologous sequences were retrieved by BLASTP searches using all 397 genes as queries (e-value: 1e−20; coverage cutoff: 0.5), (3) In three rounds, gene trees were constructed and carefully inspected in order to detect and remove putative paralogs and contaminants. For that, sequences were aligned with MAFFT v. 7.310[69] using either the -auto option (first round) or MAFFT L-INS-i with default settings (second and third round). Ambiguously aligned positions were filtered using trimAL v. 1.4[70] with a gap threshold of 0.8 (all three rounds), followed by maximum likelihood (ML) single-gene tree reconstruction with either FastTree v. 2.1.10[71] using -lg -gamma plus options for more accurate performances (first round) or RAxML v. 8.2.10[72] with PROTGAMMALGF and 100 rapid bootstrap searches (second and third round). To facilitate the detection of contaminants and paralogs, all taxa were renamed following NCBI's taxonomy (manually refined—the custom taxonomy is available in Supplementary Fig. 1; Supplementary Data 2) and colour-coded using in-house

scripts. Multiple copies from the same taxon were assigned to a unique colour allowing to more easily detect contaminants and paralogs in FigTree v. 1.4.3 (http://tree.bio.ed.ac.uk/software/figtree/). Sequences of taxa frequently observed to be nested in unrelated groups, sharing a branch with typically the same unrelated taxon in most single-gene trees, were identified as contaminants. Copies of taxa branching at unexpected positions, mostly as sister to certain clades, were identified as paralogs. In cases of very recent gene duplications, characterised by two or more paralogs of the same taxon, those with the longest branches were removed in order to minimise the chance of systematic errors caused by long-branch attraction. After the three rounds of gene tree inspection, we discarded 77 genes (74 out of the 134 genes added from Brown et al.[65]) due to suspicious clustering of major groups (e.g. duplication of the entire Sar clade in FTSJ1). The resulting dataset comprised 320 genes and 733 eukaryote taxa with ≥5% data; Supplementary Data 2.

For each curated gene, sequence stretches without clear homology (e.g. poor quality stretches of amino acids, or leftover untranslated regions) were removed with PREQUAL v. 1.01[73] employing a posterior probability threshold of 0.95 (ignoring some fast-evolving taxa). Sequences were aligned using MAFFT G-INS-i with a variable scoring matrix to avoid over-alignment (–unalignlevel 0.6) and trimmed with BMGE v. 1.12[74] using -g 0.2, -b 5, -m BLOSUM75 parameters. Partial sequences belonging to the same taxon that did not show evidence for paralogy or contamination on the gene trees were merged. All 320 trimmed gene alignments were concatenated with SCaFoS v. 1.25[75] into a supermatrix of 733 taxa and 62,723 aligned amino acid positions (62,552 distinct patterns; gaps and undetermined/missing character states: ~35%; Supplementary Data 2). This dataset was subjected to ML analysis in IQ-TREE[76] with the site-homogeneous model LG + G + F and ultrafast bootstrap approximation (UFBoot[77]; 1000 replicates) employing the -bb and -bnni flags (IQ-TREE versions 1.6.3 to 1.6.9 have been used in this study). This large tree (Supplementary Fig. 1) was used to select a reduced taxon-sampling that maintained phylogenetic diversity but allowed downstream analyses with more sophisticated models. The taxa selection aimed to (1) retain all major lineages of eukaryotes, (2) preferentially keep slowly-evolving (shorter branches) representatives, (3) preferentially discard taxa with more missing data and (4) allow the precise taxonomic placement of fossil calibrations. In order to increase sequence coverage, some monophyletic strains or species/genera complexes were combined to form a chimeric operational taxonomic unit (OTU; Supplementary Data 2). This strategy led to a dataset containing 136 OTUs, for which the raw sequences were again filtered, aligned, trimmed and concatenated as described above, forming a supermatrix with 73,460 aligned amino acid positions (71,540 distinct patterns; gaps and undetermined/missing character states: ~17%; Supplementary Data 2). Finally, we created a taxon-subset of the 136-OTU dataset containing 63 OTUs with 73,460 aligned amino acid positions (67,700 distinct patterns; gaps and undetermined/missing character states: ~13%; Supplementary Data 2), to be able to obtain chains convergence with the computationally very demanding CAT + GTR + G model.

**Phylogenomic analyses**. The reduced dataset (136 OTUs) was subjected to ML analysis in IQ-TREE[76] using the best-fit site-heterogeneous model LG + C60 + G + F with the PMSF approach to calculate non-parametric bootstrap support (100 replicates; Supplementary Fig. 2). This dataset was also analysed under the Multi Species Coalescent (MSC) approach implemented in ASTRAL-III[78] to account for incomplete lineage sorting (here, taxa were not combined into OTUs). In order to improve the phylogenetic signal in the single-gene trees used in ASTRAL, a partial filtering method (i.e. Divvier[79] using the -partial option) was applied followed by trimming of highly incomplete positions with trimAL (-gt 0.05). ML single-gene trees (Supplementary Data 2) were inferred with IQ-TREE under BIC-selected models including site-homogeneous models (such as LG) and empirical profile models (C10–C60). Branch support was inferred with 1000 replicates of ultrafast bootstrap (-bbni) and branches with <10 support were collapsed. Branch support for the MSC tree (Supplementary Fig. 3) was calculated as quartet supports[80], and multilocus bootstrapping (1000 UFboot2 bootstraps per gene).

The 63-OTU dataset was analysed under the CAT + GTR + G model in PhyloBayes-MPI v. 1.8[81]. Three independent Markov Chain Monte Carlo (MCMC) chains were run for ~2600 cycles (all sampled). The initial 500 cycles were removed (as burnin) from each chain before generating a consensus tree using the bpcomp option. Global convergence was achieved in all combinations of the chains (Supplementary Fig. 4) with maxdiff reaching 0. The 63-OTU dataset was also used in posterior predictive analyses (Supplementary Table 1) to informatively select the best topology. For that, the CAT + GTR + G tree was compared to a tree obtained using PhyloBayes-MPI v. 1.8 under LG + C60 + G + F; Supplementary Fig. 5). To remove the most heterogeneous sites from the 63-OTU dataset, the script Alignment_pruner.pl (https://github.com/novigit/davinciCode/blob/master/perl) was used. The 25%, respectively, 50% of the most heterogeneous sites were removed and the trees were inferred in PhyloBayes-MPI v. 1.8 under the CAT + GTR + G model (Supplementary Fig. 6). The reference topology used in the dating analyses corresponded to an ML analyses of the 136-OTU dataset under LG + C60 + G + F but constrained by the relationships obtained in the CAT + GTR + G tree.

**EGT detection**. To evaluate the evidence for endosymbiotic transfers of marker genes in red plastid-containing lineages, gene trees for the 320 markers were

analysed using the script count_sister_taxa.py (https://github.com/Tancata/phylo/blob/master/count_sister_taxa.py), providing the ML tree and the bootstrap files as input. We labelled each sequence in each gene tree with its taxonomic group as in Fig. 2. For each clade in the ML tree to which a single taxonomic label could be assigned, this script calculates the relative frequencies with which all other clades in the tree were recovered as the closest sister group, averaging over a sample of 1000 ultrafast bootstrap trees. The rationale is that gene-specific endosymbiotic replacement can be detected as bootstrap support for a sister-group relationship between donor and recipient lineages that conflicts with other single-gene trees or the overall species tree. When the sister clade contained sequences of mixed taxonomy, the relative frequencies of taxa in the sister clade were augmented in proportion. The result is an assessment of the signal for close relationships in single-gene trees, averaged over bootstrap replicates for the entire marker gene set.

**Molecular dating**. Bayesian molecular dating was performed with MCMCTree[82] within the PAML package v. 4.9h[83], PhyloBayes-MPI v. 1.8[81] and BEAST v. 1.10.4[84]. We used a total of 33 node calibrations based on fossil evidence (retrieved April 2019; Table 1) and the tree topology of Supplementary Fig. 4. MCMCTree was used to perform a set of sensitivity analyses (two MCMC chains for each experimental condition) in order to understand the effect of different clock models (uncorrelated or autocorrelated), tree roots and prior calibration densities. A uniform birth–death tree prior was assumed and analyses were run with either (i) uncorrelated (clock = 2) or (ii) autocorrelated (clock = 3) relaxed clock models. The tree was rooted either on (i) Amorphea[85] or (ii) Discoba (Excavata)[86]. Four prior calibration distributions were tested, following dos Reis et al.[34]: (i) uniform (i.e. maxima and minima), (ii) skew-normal ($\alpha = 10$ and $\beta$ scale parameters chosen so that the 97.5% cumulative probabilities coincide with the maxima; calculated with MCMCTreeR https://github.com/PuttickMacroevolution/MCMCtreeR), and (iii) truncated-cauchy distributions with either short ($p = 0$; $c = 0.1$; pL = 0.01) or (iv) long ($p = 0$; $c = 10$; pL = 0.01) distribution tails. Skew-normal distributions represent 'literal' interpretations of the fossil record, assuming minima are close to the real node ages, whereas truncated-cauchy distributions represent more 'loose' interpretations that assume older divergences than minimum bounds[34]. In all cases, the root was calibrated using a uniform distribution 1.6–3.2 Ga. All maxima and minima were treated as soft bounds with a default 2.5% prior probability beyond their limits. MCMCTree analyses were run on the entire concatenated alignment using approximate likelihood calculations[87]. Data were analysed as a single partition under the LG + G model. The prior on the mean (or ancestral) rates ('rgene_gamma') were set as diffuse gamma Dirichlet priors indicating severe among-lineage rate heterogeneity and mean rates of 0.02625 and 0.0275 amino acid replacements site$^{-1}$ $10^8$ Myr$^{-1}$, for, respectively, the Amorphea ($\alpha = 2$, $\beta = 72.73$) and Excavata ($\alpha = 2$, $\beta = 76.19$) roots. The average rate was calculated as mean root-to-tip paths on the corresponding ML trees with the two roots. The rate drift parameter ('sigma2_gamma') was set to indicate considerable rate heterogeneity across lineages ($\alpha = 2$, $\beta = 2$). A 100 Ma time unit was assumed. Two independent MCMC chains were run for each analysis, each consisting of 20.2 million generations, of which the first 200,000 were excluded as burnin. Convergence of chains was checked a posteriori using Tracer v. 1.7.1[88] and all parameters reached ESS >200. To test the effect of the oldest eukaryote fossil calibration (in red algae), additional analyses were performed under autocorrelated and uncorrelated clock models (assuming uniform prior calibrations and the Amorphea root). A total of 36 analyses were run, corresponding to all possible combinations of two root positions, two clock models, four calibration distributions and two MCMC chains per experimental condition, and four analyses after the exclusion of the Rhodophyta fossil.

For computational tractability, PhyloBayes and BEAST were run on a subset of the ten most clock-like genes (selected with SortaDate[89]). PhyloBayes analyses were run under (i) uncorrelated and (ii) autocorrelated relaxed clock models and using either the (i) site-homogeneous LG + G or (ii) site-heterogeneous CAT + GTR + G models. In this case, calibrations were set as uniform priors with soft bounds and assumed a birth-death tree prior and the same tree topology (rooted on Amorphea). Two independent MCMC chains were run until convergence, assessed by PhyloBayes' built-in tools (bpcomp and tracecomp). For comparative purposes, we run an additional BEAST analysis with similar parameterisations (among those available in BEAST): the uncorrelated relaxed clock, a fixed tree topology rooted on Amorphea, uniform prior calibrations, a Yule tree prior and the LG + G evolutionary model. Two independent MCMC chains were run for 200 million generations, the first 10% being discarded as burnin. Convergence of chains was checked with Tracer and all parameters reached ESS >200. CorrTest[35] was used to test the autocorrelation of branch lengths.

**Reporting summary**. Further information on research design is available in the Nature Research Reporting Summary linked to this article.

## Data availability
All data needed to evaluate the conclusions of this study are present in the paper, the Supplementary Information and the Supplementary Data. Raw sequence data are available under the following web-links: https://ensemblgenomes.org, https://imicrobe.us/#/projects/104, https://ncbi.nlm.nih.gov, https://onekp.com/samples/list.php, https://doi.org/10.6084/m9.figshare.12417881.v2.

## Code availability
count_sister_taxa.py is available under: https://github.com/Tancata/phylo/blob/master/count_sister_taxa.py.

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

## Acknowledgements

This work was supported by a grant from Science for Life Laboratory available to F.B., which covered the salary of J.F.H.S. and I.I. I.I. acknowledges support from the Spanish Ministry of Economy and Competitiveness (MINECO) (Juan de la Cierva fellowship IJCI-2016- 29566) and the European Research Council (Grant Agreement No. 852725; ERC-StG 'TerreStriAL' to Jan de Vries, University of Gottingen). T.A.W. was supported by a Royal Society University Research Fellowship and NERC Grant NE/P00251X/1. Computations were performed on resources provided by the Swedish National Infrastructure for Computing (SNIC) at Uppsala Multidisciplinary Center for Advanced Computational Science (UPPMAX) under Projects 2017-7-65, 2017-7-355, 2018-3-147, 2018-3-288, 2018-8-187, 2018-8-192 and 2019-3-305.

## Author contributions

F.B. conceived and supervised the project. J.F.H.S. assembled and curated the data, and performed phylogenomic analyses. T.A.W. analysed data under the Multispecies Coalescent model and tested for endosymbiotic gene transfer. J.F.H.S. compiled the fossil calibrations, and I.I. designed and performed molecular dating analyses. All authors drafted the manuscript and read and approved the final version.

## Funding

## Competing interests

The authors declare no competing interests.
