## [Peer Review File · Nature Communications]

Reviewers' Comments:

Reviewer #1:

Remarks to the Author:

This represents an interesting and noteworthy investigation into the origin of red plastids utilizing molecular clock approaches. Using an expanded set of phylogenetic markers plus taxon sampling, Cryptista is shown to be related to Archaeplastida. This topology is not compatible with the hypothesis that red plastid-containing algae originated from a single plastid generating endosymbiotic event involving a red algal symbiont. Further, based on time calibrated phylogeny, the authors identified two models proposed previously by others that are compatible with their results. However, there is room for improvement in data interpretation. The conclusions could be strengthened by systematically testing possible scenarios. Also, integration of some key information (e.g. rpl36 and other plastid related phylogenetic, biochemical features) in the discussion is needed. Specific comments are as follows:

1. Serial plastid models: the authors opted to test existing hypotheses out there regarding the evolution of red plastid-bearing eukaryotic groups. Of those, two models (presented in Fig 1) fit with their time-calibrated phylogeny. Assuming what have been proposed are not exhaustive, it would be helpful and quite relevant to look into any other possible scenarios that are compatible with the results.
2. The manuscript can be strengthened by incorporating other relevant known details regarding the evolution of red plastids (phylogeny, biochemistry, morphology, etc.) in the discussion. For example, rpl36 represents a shared lateral gene transfer event that unites haptophyte and cryptophyte plastids, but it is not discussed here.
3. Concatenated analyses could be misleading as few markers are able to skew the results based on hundreds of genes. Given this, it is useful to examine how the main topology presented, especially related to red plastid-containing groups, are supported by individual tree analyses.
4. Data curation: details on identification and removal of paralogs and contaminant sequences are missing. For example, what methods were used for identifying contaminants? What is the extent of read contamination in each data set, especially given that many protist transcriptome data are not clean?
5. "Predatory behaviours have been demonstrated in... non-photosynthetic direct relatives of red algae, suggesting that mixotrophy was a key intermediate stage in early evolution of Archaeplastida": Rhodelphis is not a mixotroph, but rather the only obligate predatory flagellate known to date within Archaeplastida.
6. Fig 2: curious to know why picozoan is removed in this representative tree. Where does it go when included?
7. "these nutrient-poor environments could not sustain a radiation of eukaryotic algae": I do not follow this logic given that microbial taxon diversity tends to be high in oligotrophic freshwater or marine waters. In fact, eutrophic environments, especially those impacted by human activities, are characterized by low biodiversity despite having a higher total algal biomass.
8. Supplementary data: the authors provide various supplementary files, including sequence alignments, which will be useful for other researchers in the field. It is going to be of good value to the community if individual tree files (in pdf or jpeg) are included as well.

Reviewer #2:

Remarks to the Author:

This investigation tests the validity of recent models for how plastids descended from red algae could have moved via serial endosymbioses through eukaryotic lineages. Specifically, the authors generate a phylogenetic hypothesis for broad scale eukaryotic relationships using standard and rigorous computational approaches, then overlay inferred times of origin of relevant algal lineages using molecular clock calibrations. The goal is to determine whether various models are plausible given

when the algal groups in question originated.

Using the guidelines provided to reviewers, the following are my observations on this paper.

1. What are the noteworthy results?

Although the results do not provide direct or explicit support for a specific model of serial endosymbiosis, they do reject a null hypothesis that proposed serial transfers are not possible based on when each photosynthetic lineage emerged. The overall results provide somewhat greater support for one model than for others, and that should help direct researchers toward further fruitful investigations. As someone who has argued for more explicit hypothesis testing in eukaryotic phylogenomics, as opposed to continuous debates over disagreements among tree topologies, I find this study refreshing and likely significant for the fields of eukaryotic and algal evolution.

2. Will the work be of significance to the field and related fields? How does it compare to the established literature? (If the work is not original, please provide relevant references)

Certainly, there is no shortage of similar broad scale phylogenetic investigations of eukaryotic evolution, nor of attempts to date the origins of major eukaryotic lineages using molecular clocks. To my knowledge, however, this is the first attempt to test explicit predictions of recent models of serial plastid endosymbioses using these approaches. At least in some cases, serial plastid transfers have been proposed based upon entirely different methodological approaches. As such, the findings of this study represent an independent test of several controversial and mutually exclusive models of plastid evolution.

3. Does the work support the conclusions and claims, or is additional evidence needed? Are there any flaws in the data analysis, interpretation and conclusions? Do these prohibit publication or require revision?

The results support the authors' conclusions, which are neither over- nor inappropriately interpreted. I do have several comments regarding potential complications regarding the overall phylogenetic methodology employed. I will say, up front, that these are not specific criticisms of this investigation; rather, they reflect what, in my view, are serious challenges that phylogenomic researchers need to begin to address more generally. I don't suggest this paper needs to be the one to do that. Nevertheless, it would be worthwhile for these authors to consider discussing, or at least acknowledging, some issues with implications for their results.

Let me begin with general considerations, then move to specifics for this study. Many major conclusions from molecular phylogenetic and phylogenomic research were developed before an recognition of the prevalence and impacts of endosymbioses on eukaryotic genomes became clear. As such, there are hardened conclusions and basic methodologies now baked into the field that don't necessarily address complications of complex patterns of eukaryotic to eukaryotic endosymbioses. In my view, these methodologies, regardless of the care and rigor employed, are inadequate for addressing evolutionary problems wherein some unknown fraction of sequences analyzed have a persistent, reticulate history that does not track the evolutionary pattern of the genomes in which they reside. I think the assumption has been that the overall phylogenetic signal from a genome will come from genealogical rather than horizontal descent. Thus, with data sets composed of many genes, genealogical signal will swamp out reticulate signal and produce a tree that recapitulates the evolutionary history of cellular organisms. This likely is true for random horizontal transfers, but what about the impacts of unknown but persistent, directional signal from secondary or serial plastid endosymbioses? Many of the deeper branches in eukaryotic trees are difficult to recover consistently, if at all. Persistent, directional but conflicting phylogenetic signals could have major impacts in these cases. Intra-genome phylogenetic conflicts have been known from the early days of molecular phylogenies but largely have been waived away as a nuisance that can be overcome with large enough data sets. It is relatively easy to factor out genes of bacterial origin; however, movements and

replacements among eukaryotic homologs are harder to detect. Given the evidence for large-scale endosymbiotic movement of genes, I think traditional assumptions are more inadequate than ever. Because the framework of this investigation relates directly to the possibility of large-scale horizontal movements of genes, I think there are several specific issues these authors should consider discussing. First, can it be ruled out that EGT is impacting the basic tree topology? For example, the authors discuss conflicting positions of haptophytes and cryptophytes in various studies, depending on the data and approaches used. Could these reflect, at least in part, real intra-genomic conflicts that are relevant to the current results? In the final consensus tree in this study, cryptophytes and their relatives are recovered as sister to the Archaeplastida (as they have been in some other analyses). Given that cryptophytes clearly adopted an Archaeplastidal endosymbiont, couldn't their position on the tree be impacted by the presence of unidentified red algal gene replacements? How do the authors rule out this possibility? Such a hypothesis is consistent with the phylogenetic results of this and other studies but, to my knowledge, has not been investigated carefully much less ruled out. Likewise, if haptophyte genomes have been impacted by an influx of, for example, chromophyte genes, couldn't that persistent phylogenetic signal pull them toward stramenopiles on the molecular tree of life? I think the potential impacts extend beyond just the tree topology. For example, what are the likely effects of unidentified genome mixing on inferred internode lengths? Wouldn't real intra-genomic conflicts tend to expand basal internodes, thereby generally pushing back inferred times of origins? The authors spend a lot of text discussing the implications of long periods between the origins of these groups and their ecological expansions. Wouldn't it be prudent to discuss whether long internodes leading to algal groups could be artifacts of current computational approaches to analyzing chimeric genomes? Note that a general compression of deeper internodes would not be inconsistent with the overall conclusions of the study, nor with the authors' geological and ecological arguments for why major eukaryotic algal groups expanded when they did.

I do not mean to suggest that these authors solve what remains a thorny problem in phylogenomics. I do think they could discuss such issues in their overall interpretations of results. In the long term, I don't believe we will come to a clear understanding of eukaryotic and plastid evolution until intra-genome phylogenetic conflicts are addressed directly, and within frameworks that test endosymbiotic models explicitly. This study addresses the latter but not the former. I will say that, given the rigor and capabilities of these researchers, I would love to see them turn their attention to those kinds of issues in the future.

4. Is the methodology sound? Does the work meet the expected standards in your field? Is there enough detail provided in the methods for the work to be reproduced?

The methodologies appear sound. No comments or suggestions here.

John W. Stiller

Reviewer #3:

Remarks to the Author:

Review of Nature Communications manuscript NCOMMS-20-33920

In this interesting manuscript, Jürgen Strasser, Iker Irisarri, Tom Williams, and Fabien Burki focused on the timescale for the origin of plastids derived from red algae. First of all, they inferred a well-resolved phylogenetic tree of eukaryotes from a 320-protein dataset. For example, they recovered a robust sister group relationship of Cryptista and Archaeplastida. Then, they estimated divergence times between and within major eukaryotic clades, and they convincingly validated their results through the use of different sequence evolution models and relaxed molecular clocks. Based on the chronograms, they use the inescapable assumption that for any endosymbiotic relationship to be established, the endosymbiont and the host must live at the same time (and in the same place) in order to interact. Eventually, they conclude that the hypothesis of serial endosymbioses is

chronologically possible, allowing red plastids to have been passed between distantly related hosts. The serial transfers may have all taken place in a relatively short time window between 650 and 1,079 million years, from the initial secondary endosymbiosis in a stem cryptophyte to the engulfment of plastids in the ancestors of all modern-day red algal plastid-containing lineages.

This manuscript is clearly written, with an appropriate length. The results are novel, convincing, and not oversold. Sufficient methodological details are provided, allowing reproducing the analyses. As the claims are appropriately discussed in the context of a fair treatment of previous literature, they contribute to a better understanding of the evolution of complex red algal-derived plastids (in this context, Figure 1 is of prime importance). Because this manuscript shows how a phylogenetic timeframe can allow testing for different scenarios of symbioses, it will be of interest to researchers in the field, and influence their thinking as well.

I have the following major and minor comments — for example about statistical analyses and methodological investigations — that I hope the authors will consider in a revised, strengthened, future version of their manuscript.

MAJOR POINTS

1. Lines 116-119. We know that taxon sampling is of paramount importance to infer phylogenies, especially for deep cladogenetic events. Therefore, it is important to argue why the pruning of the 733-OTU dataset to obtain the 136-OTU and 63-OTU reduced datasets did not significantly impact the topologies inferred. In other words, what is the effect on the tree inference of the taxonomic diversity, the fast / slow branches, the missing data, and the calibration information (cf. L. 418-420)? With these criteria, is there any trade-off problem to be solved, e.g., keeping a longer branch and/or a superprotein with more missing data provided it brings additional taxonomic representatives or calibration constraints? And what about the possibility of sorting / discarding individual protein alignments according to their average rate of amino acid replacements?
2. L195-198: To summarize all estimated node ages, the mean is provided with respect to different calibration priors. However, if these age estimates are not Gaussian-distributed, the mean is not a valid statistical descriptor, and it would be better to use the median with quantiles. These quantiles would also allow fair comparisons among results obtained from different approaches. For example, when it is stated that "The younger ages inferred under the autocorrelated clock model were more apparent when uniform calibrations were applied (mean ages of 904 vs. 947 Ma) and this trend was inverted in the case of skew normal distributions (913 vs. 896 Ma)", there is no statistical background to support this claim because of the partial overlap of credibility intervals.
3. L798. In the calibrations used for dating the eukaryote tree of life, a minimum bound at 1,600 Ma (Paleo/Meso-proterozoic eon) has been set for Rhodophyta. Then, the authors conclude that both the fossil record and molecular clock inferences support a Paleoproterozoic origin of primary plastids, and an early Mesoproterozoic origin of red algae (L. 299-300). The possibility of circularity in these results should however be evaluated. For example, under a leave-one-out procedure, is there any change in the node age estimates if the Rhodophyta calibration is removed? And given the calibrating information constrained on the other nodes, would we again infer early Mesoproterozoic estimates for Rhodophyta? More generally, what is the sensitivity of the molecular clock analyses with respect to the various calibration constraints?
4. The reader is not informed about data availability (alignments, trees, chronograms).

MINOR POINTS

1. Lines 127-128: Trees inferred are said to be "in good overall agreement" with current knowledge of the broad eukaryote evolution. Similarly, one tree is "globally consistent" with others (L. 142). These are vague wordings. Actually, which percentage of the total number of nodes is compatible with a given reference topology?
2. L146-147: It is written, "topological differences observed are derived from the chosen evolutionary model and not from the use of Bayesian or ML inference." Why do you expect differences from these two different probability approaches? Please expand on this matter, and/or provide a reference. If you are thinking about prior issues, state it clearly.
3. L377, and throughout the text: the authors mention "genes" for some analyses, whereas I assume that "proteins" have been analyzed. Please clarify what is the primary source of information (nucleotides or amino acids).
4. L125. As the site heterogeneous CAT+GTR model can also be run without Gamma distribution, the simpler "catgtr" name is rather confusing. Why not using "catgtrg", or more simply keeping CAT+GTR+G?
5. L441. As a character is a column in a sequence alignment, I assume that the "undetermined characters" are "undetermined / missing character states".
6. Even if the manuscript is centered on time-calibrated phylogenies, it might be worth providing the reader with a supplementary figure showing the highest-likelihood or maximum probability tree with branch lengths. This would allow appreciating the extent of evolutionary rate variations among clades.
7. Figure 1 is very important to understand the scientific context of this manuscript. For this reason, it would help to place the primary endosymbiosis event leading to primary plastids in photosynthetic eukaryotes.

Reviewer #1 (Remarks to the Author):

This represents an interesting and noteworthy investigation into the origin of red plastids utilizing molecular clock approaches. Using an expanded set of phylogenetic markers plus taxon sampling, Cryptista is shown to be related to Archaeplastida. This topology is not compatible with the hypothesis that red plastid-containing algae originated from a single plastid generating endosymbiotic event involving a red algal symbiont. Further, based on time calibrated phylogeny, the authors identified two models proposed previously by others that are compatible with their results. However, there is room for improvement in data interpretation. The conclusions could be strengthened by systematically testing possible scenarios. Also, integration of some key information (e.g. rpl36 and other plastid related phylogenetic, biochemical features) in the discussion is needed. Specific comments are as follows:

1. Serial plastid models: the authors opted to test existing hypotheses out there regarding the evolution of red plastid-bearing eukaryotic groups. Of those, two models (presented in Fig 1) fit with their time-calibrated phylogeny. Assuming what have been proposed are not exhaustive, it would be helpful and quite relevant to look into any other possible scenarios that are compatible with the results.

> We thank the reviewer for this suggestion. As indicated by the reviewer and in our manuscript, including in the figure legend (fig 1), we have limited the range of possibilities to the models that are both (i) published and (ii) compatible with our results: “*Serial plastid endosymbioses models as proposed by Stiller et al.¹⁸ (left) and Bodyl et al.⁴⁴ (right). The tree topology shown here is based on the results obtained in our study. Further models have been suggested but are not compatible with this topology^{15,87,88}”.* This was done for clarity and with the aim to build on previous work. Indeed, given that the chronology of plastid acquisition shown in this study does not rule out any specific pairing of plastid donor/recipient (except perhaps the secondary acquisition of red plastid by ochrophytes, which we discuss), we would need to discuss an unmanageable number of permutations if we attempted to look into any compatible scenarios. Therefore, we respectfully prefer to maintain our decision to focus on scenarios that have been discussed in the literature and are compatible with our topology (which incidentally allows us to rule out a number of models that are not compatible with our topology). Note however, that two additional models are referred to in the discussion, those of Bodyl et al. 2009 and Bodyl 2018.

2. The manuscript can be strengthened by incorporating other relevant known details regarding the evolution of red plastids (phylogeny, biochemistry, morphology, etc.) in the discussion. For example, rpl36 represents a shared lateral gene transfer event that unites haptophyte and cryptophyte plastids, but it is not discussed here.

> This is a very valuable comment and we have accordingly added a paragraph in the discussion covering two very important characters often used as strong support of the chromalveolate hypothesis or specific association among the chromalveolate lineages. These characters are: (i) SELMA, part of the translocon machinery, which is uniquely used to cross the second outermost membrane in red complex plastids, and (ii) as suggested by the reviewer the bacterial horizontally transferred rpl36 gene into the plastid genomes of cryptophytes and haptophytes. We paste the new paragraph here for easier evaluation: “*The rhodoplex hypothesis also allows to reconcile non-phylogenetic plastid data with that of their hosts. The mechanism of protein import into complex red plastids involves a unique translocation machinery known as SELMA, which is derived from the ER-associated protein degradation (ERAD) system of the red algal endosymbiont⁴². SELMA has been interpreted as a robust character supporting a single plastid origin in a chromalveolate ancestor³⁴, but the presence*

of this machinery in all algae with red plastids (with the exception of myxozoans) is also compatible with serial acquisitions⁵. The molecular components of SELMA are nucleus- or nucleomorph-encoded in all investigated organisms, which implies that these genes would have been repeatedly transferred to the host nucleus during each rhodoplex endosymbiosis. Although this may appear less likely than a single origin of SELMA followed by vertical inheritance, it is worth noting that a similar mechanism of independent nuclear relocalisations of homologs of the TIC/TOC protein import machinery took place in the green lineages chlorarachniophytes and euglenids, as well as in the red algal plastid-containing lineages⁹. Thus, the possibility exists that SELMA has also been successively reestablished during the process of serial endosymbiosis. Another plastid character that is in apparent contradiction with our host-derived topology is the horizontally transferred bacterial *rpl36* gene into the plastid genomes of haptophytes and cryptophytes⁴³. Here again, the rhodoplex hypothesis is compatible with the existence of a specific link between the haptophyte and cryptophyte plastids but not between the hosts. In fact, this is an explicit possibility in the model of Bodyl et al.^{44,45}, which proposed that plastids were transferred twice from cryptophytes: once to ochrophytes before the *rpl36* lateral gene transfer and then to haptophytes after the *rpl36* replacement by a bacterial homolog.”

3. Concatenated analyses could be misleading as few markers are able to skew the results based on hundreds of genes. Given this, it is useful to examine how the main topology presented, especially related to red plastid-containing groups, are supported by individual tree analyses.

> We agree with the reviewer that especially endosymbiotic gene transfers (EGT) could be problematic to infer the species tree of the focal groups of organisms in this study, if there indeed was a directional undetected reticulation in the data that we used. To verify this and hopefully satisfy the reviewer’s concern, we added a new analysis that systematically evaluated the strength of support for gene transfer events associated with plastid acquisition. This analysis is described in its own subsection in the Methods: “**EGT detection.** To evaluate the evidence for endosymbiotic transfers of marker genes in red plastid-containing lineages, gene trees for the 320 markers were analysed using the script `count_sister_taxa.py` (https://github.com/Tancata/phylo/blob/master/count_sister_taxa.py), providing the ML tree and the bootstrap files as input. We labelled each sequence in each gene tree with its taxonomic group as in Fig. 2. For each clade in the ML tree to which a single taxonomic label could be assigned, this script calculates the relative frequencies with which all other clades in the tree were recovered as the closest sister group, averaging over a sample of 1,000 ultrafast bootstrap trees. The rationale is that gene-specific endosymbiotic replacement can be detected as bootstrap support for a sister group relationship between donor and recipient lineages that conflicts with other single gene trees or the overall species tree. When the sister clade contained sequences of mixed taxonomy, the relative frequencies of taxa in the sister clade were augmented in proportion. The result is an assessment of the signal for close relationships in single gene trees, averaged over bootstrap replicates for the entire marker gene set.”

We also added an interpretation of this in the results, which includes a link to a new figure: “Finally, we evaluated whether our selected phylogenetic markers displayed signal resulting from potential endosymbiotic gene transfers (EGTs) between the endosymbiont and host genomes during plastid establishment⁹. Relationships among algae might be affected by EGT, which, if undetected, would distort the species tree and compromise our efforts to test hypotheses of red plastid spread by reference to the host phylogeny. For example, the inferred sister relationship between Cryptista and Archaeplastida could be an artefact due to the replacement of host cryptophyte genes by homologs from the red algal endosymbiont. We systematically evaluated bootstrap support for sister-group relationships between each red

plastid-containing lineage and all other eukaryotic taxonomic groups for each of the 320 marker genes independently (Material and Methods). Importantly, this analysis provided no evidence for horizontal acquisition of any marker genes during evolution, as we did not detect dominant non-vertical signals for any group. That is, bootstrap support from single genes was either equivocal for deep relationships, or favoured the lineage that would be expected based on prior knowledge of species relationships (Fig. 2)."

Finally, we would like to point out to the Astral analysis already included in the submitted version of the manuscript, which is compatible under a Multispecies Coalescent model and thus takes into account variability at the single-gene level; it is worth noting that these kinds of models have generally not been used in other global phylogenomic analyses of eukaryotes.

4. Data curation: details on identification and removal of paralogs and contaminant sequences are missing. For example, what methods were used for identifying contaminants? What is the extent of read contamination in each data set, especially given that many protist transcriptome data are not clean?

> Contaminants and paralogs have been manually identified and removed following three rounds of careful gene tree examination. Unfortunately, we are not aware of available automatic software that can perform a similar level of detailed curation. Resulting from the manual curation, the extent of read contamination in each input dataset was not traced and cannot be reported here. However, to address the reviewer's request on how contaminants and paralogs have been identified, we added the missing information to the first paragraph of the Methods, which now reads: "*Sequences of taxa frequently observed to be nested in unrelated groups, sharing a branch with typically the same unrelated taxon in most single gene trees, were identified as contaminants. Copies of taxa branching at unexpected positions, mostly as sister to certain clades, were identified as paralogs. In cases of very recent gene duplications, characterised by two or more paralogs of the same taxon, those with the longest branches were removed.*"

5. "Predatory behaviours have been demonstrated in... non-photosynthetic direct relatives of red algae, suggesting that mixotrophy was a key intermediate stage in early evolution of Archaeplastida": Rhodelphis is not a mixotroph, but rather the only obligate predatory flagellate known to date within Archaeplastida.

> We apologise for the confusion, we have reworked this sentence and hope this is clearer now: "*Predatory behaviours have been demonstrated in green algae and non-photosynthetic direct relatives of red algae, indicating that phagotrophy persisted alongside phototrophy for long evolutionary times in Archaeplastida and that mixotrophy was a key intermediate stage in early evolution of plastids*^{58,59}."

6. Fig 2: curious to know why picozoan is removed in this representative tree. Where does it go when included?

> That is a good question! In our analysis of the full dataset, Picozoa branch as sister to the Telonemids, however this position is unsupported (Supplementary Fig. 1). Picozoa can be considered a rogue taxon; in all our analyses when included, it disturbs deep nodes of the eukaryotic tree due to its unstable position. We think that this is due to the highly incomplete coverage of the available data (82% missing data in our 320-gene dataset), and representing only one branch of this phylum. For these reasons, we have removed it from the analyses aimed at determining the eukaryote species tree. More Picozoa data are crucially needed.

7. "these nutrient-poor environments could not sustain a radiation of eukaryotic algae": I do not follow this logic given that microbial taxon diversity tends to be high in oligotrophic freshwater or marine waters. In fact, eutrophic environments, especially those impacted by

human activities, are characterized by low biodiversity despite having a higher total algal biomass.

> We agree that this was a confusing statement. We have now removed the quoted part from the discussion and hope that this makes our speculative reasoning clearer.

8. Supplementary data: the authors provide various supplementary files, including sequence alignments, which will be useful for other researchers in the field. It is going to be of good value to the community if individual tree files (in pdf or jpeg) are included as well.

> We agree and added a file containing all single gene trees to Supplementary Data 2.

Reviewer #2 (Remarks to the Author):

This investigation tests the validity of recent models for how plastids descended from red algae could have moved via serial endosymbioses through eukaryotic lineages. Specifically, the authors generate a phylogenetic hypothesis for broad scale eukaryotic relationships using standard and rigorous computational approaches, then overlay inferred times of origin of relevant algal lineages using molecular clock calibrations. The goal is to determine whether various models are plausible given when the algal groups in question originated.

Using the guidelines provided to reviewers, the following are my observations on this paper.

1. What are the noteworthy results?

Although the results do not provide direct or explicit support for a specific model of serial endosymbiosis, they do reject a null hypothesis that proposed serial transfers are not possible based on when each photosynthetic lineage emerged. The overall results provide somewhat greater support for one model than for others, and that should help direct researchers toward further fruitful investigations. As someone who has argued for more explicit hypothesis testing in eukaryotic phylogenomics, as opposed to continuous debates over disagreements among tree topologies, I find this study refreshing and likely significant for the fields of eukaryotic and algal evolution.

2. Will the work be of significance to the field and related fields? How does it compare to the established literature? (If the work is not original, please provide relevant references)

Certainly, there is no shortage of similar broad scale phylogenetic investigations of eukaryotic evolution, nor of attempts to date the origins of major eukaryotic lineages using molecular clocks. To my knowledge, however, this is the first attempt to test explicit predictions of recent models of serial plastid endosymbioses using these approaches. At least in some cases, serial plastid transfers have been proposed based upon entirely different methodological approaches. As such, the findings of this study represent an independent test of several controversial and mutually exclusive models of plastid evolution.

3. Does the work support the conclusions and claims, or is additional evidence needed? Are there any flaws in the data analysis, interpretation and conclusions? Do these prohibit publication or require revision?

The results support the authors' conclusions, which are neither over- nor inappropriately interpreted. I do have several comments regarding potential complications regarding the overall phylogenetic methodology employed. I will say, up front, that these are not specific

criticisms of this investigation; rather, they reflect what, in my view, are serious challenges that phylogenomic researchers need to begin to address more generally. I don't suggest this paper needs to be the one to do that. Nevertheless, it would be worthwhile for these authors to consider discussing, or at least acknowledging, some issues with implications for their results.

> We thank Dr. Stiller for his positive and constructive comments on our manuscript. We note that all points raised below are framed as recommendations (but all very valuable) rather than requirements. We have tried to comprehensively address Dr. Stiller's main concerns.

Let me begin with general considerations, then move to specifics for this study. Many major conclusions from molecular phylogenetic and phylogenomic research were developed before an recognition of the prevalence and impacts of endosymbioses on eukaryotic genomes became clear. As such, there are hardened conclusions and basic methodologies now baked into the field that don't necessarily address complications of complex patterns of eukaryotic to eukaryotic endosymbioses.

> We agree to some extent, especially with regards to some of the conclusions derived from cell or molecular biology of plastids, perhaps less so when it comes to a possible impact of reticulate evolution of the set of host marker genes used in phylogenomics; see below (but it is worth noting here already that this and other phylogenomic investigations of eukaryotes are based on very restricted, highly curated, sets of phylogenetic markers that show now detectable evidence of the kinds of artifacts that we could expect from pervasive and dominant endosymbiotic gene transfers).

In my view, these methodologies, regardless of the care and rigor employed, are inadequate for addressing evolutionary problems wherein some unknown fraction of sequences analyzed have a persistent, reticulate history that does not track the evolutionary pattern of the genomes in which they reside. I think the assumption has been that the overall phylogenetic signal from a genome will come from genealogical rather than horizontal descent. Thus, with data sets composed of many genes, genealogical signal will swamp out reticulate signal and produce a tree that recapitulates the evolutionary history of cellular organisms. This likely is true for random horizontal transfers, but what about the impacts of unknown but persistent, directional signal from secondary or serial plastid endosymbioses?

> Fair point, we try to address it below.

Many of the deeper branches in eukaryotic trees are difficult to recover consistently, if at all. Persistent, directional but conflicting phylogenetic signals could have major impacts in these cases. Intra-genome phylogenetic conflicts have been known from the early days of molecular phylogenies but largely have been waived away as a nuisance that can be overcome with large enough data sets. It is relatively easy to factor out genes of bacterial origin; however, movements and replacements among eukaryotic homologs are harder to detect. Given the evidence for large-scale endosymbiotic movement of genes, I think traditional assumptions are more inadequate than ever.

Because the framework of this investigation relates directly to the possibility of large-scale horizontal movements of genes, I think there are several specific issues these authors should consider discussing. First, can it be ruled out that EGT is impacting the basic tree topology?

> It is difficult to totally rule this out as a possibility due to the antiquity—thus faintness—of the signal that we are trying to recover and evaluate. However, following Dr. Stiller's concerns, which was also raised by R1 (see point 3 above), we performed a new analysis aimed at detecting such EGT signal by systematically looking at close relationships between groups of eukaryotes—in particular algae—for all our marker genes and averaged over bootstrap replicates. The output of this analysis is presented as a new figure (Fig. 2) and is discussed in the main text. Rather than reproducing here again the new text (which can be read in our

answer of point 3 of R1), we will just restate the main conclusion, which is that we could not detect any signal raising above background that could be attributed to EGT. This is actually a very interesting result, because this criticism often surfaces in one form or another when presenting eukaryote phylogenomic trees, and until now we did not have quantitative data to support our claim that these housekeeping genes that make the bulk of our dataset have most likely not been acquired by EGT (or LGT for that matter).

For example, the authors discuss conflicting positions of haptophytes and cryptophytes in various studies, depending on the data and approaches used. Could these reflect, at least in part, real intra-genomic conflicts that are relevant to the current results? In the final consensus tree in this study, cryptophytes and their relatives are recovered as sister to the Archaeplastida (as they have been in some other analyses). Given that cryptophytes clearly adopted an Archaeplastidal endosymbiont, couldn't their position on the tree be impacted by the presence of unidentified red algal gene replacements? How do the authors rule out this possibility? Such a hypothesis is consistent with the phylogenetic results of this and other studies but, to my knowledge, has not been investigated carefully much less ruled out. Likewise, if haptophyte genomes have been impacted by an influx of, for example, chromophyte genes, couldn't that persistent phylogenetic signal pull them toward stramenopiles on the molecular tree of life?

> All of the above questions are very justified. However, as discussed above we do not see in our set of marker genes any indication that the stated relationships could be artefactual. Instead, we think that the new analysis performed as part of this revision—and the new text in the manuscript—reinforces our conclusion that the tree presented represents the species tree.

I think the potential impacts extend beyond just the tree topology. For example, what are the likely effects of unidentified genome mixing on inferred internode lengths? Wouldn't real intra-genomic conflicts tend to expand basal internodes, thereby generally pushing back inferred times of origins? The authors spend a lot of text discussing the implications of long periods between the origins of these groups and their ecological expansions. Wouldn't it be prudent to discuss whether long internodes leading to algal groups could be artifacts of current computational approaches to analyzing chimeric genomes? Note that a general compression of deeper internodes would not be inconsistent with the overall conclusions of the study, nor with the authors' geological and ecological arguments for why major eukaryotic algal groups expanded when they did.

> We again thank Dr. Stiller for raising this point. Although as discussed above, we did not detect evidence of genome mixing in our set of phylogenetic markers, the process of endosymbiosis undoubtedly impacted the molecular evolution of algae. In principle the substitution rate might be higher in algae due for example to shift in ecological mode/pop gen environment/"way of being" after plastid acquisition; or endogenous molecular evolution, e.g. even though the host cell genes weren't replaced, they had to adapt to function together with the new compartment, and many genes probably changed function/expression level leading to an excess number of substitutions. Even though we don't see any obvious evidence for these effects, it is difficult to exclude them entirely, and more calibrations around the key nodes (e.g. plastid-bearing vs. ancestrally non-photosynthetic cryptist lineages) might help to resolve the matter. In order to acknowledge this while keeping the discussion in focus, we added a sentence in the discussion: "*This long inferred period could be due, at least in part, to processes difficult to model, such as an early burst of evolutionary changes during endosymbiotic integration⁵⁷, or the general mosaic nature of algal genomes⁹.*"

I do not mean to suggest that these authors solve what remains a thorny problem in phylogenomics. I do think they could discuss such issues in their overall interpretations of results.

> We agree with Dr. Stiller and thank him for these very valuable discussion points. We believe that we have addressed them, at least the most pressing ones.

In the long term, I don't believe we will come to a clear understanding of eukaryotic and plastid evolution until intra-genome phylogenetic conflicts are addressed directly, and within frameworks that test endosymbiotic models explicitly. This study addresses the latter but not the former. I will say that, given the rigor and capabilities of these researchers, I would love to see them turn their attention to those kinds of issues in the future.

4. Is the methodology sound? Does the work meet the expected standards in your field? Is there enough detail provided in the methods for the work to be reproduced?

The methodologies appear sound. No comments or suggestions here.

John W. Stiller

Reviewer #3 (Remarks to the Author):

Review of Nature Communications manuscript NCOMMS-20-33920

In this interesting manuscript, Jürgen Strasser, Iker Irisarri, Tom Williams, and Fabien Burki focused on the timescale for the origin of plastids derived from red algae. First of all, they inferred a well-resolved phylogenetic tree of eukaryotes from a 320-protein dataset. For example, they recovered a robust sister group relationship of Cryptista and Archaeplastida. Then, they estimated divergence times between and within major eukaryotic clades, and they convincingly validated their results through the use of different sequence evolution models and relaxed molecular clocks. Based on the chronograms, they use the inescapable assumption that for any endosymbiotic relationship to be established, the endosymbiont and the host must live at the same time (and in the same place) in order to interact. Eventually, they conclude that the hypothesis of serial endosymbioses is chronologically possible, allowing red plastids to have been passed between distantly related hosts. The serial transfers may have all taken place in a relatively short time window between 650 and 1,079 million years, from the initial secondary endosymbiosis in a stem cryptophyte to the engulfment of plastids in the ancestors of all modern-day red algal plastid-containing lineages.

This manuscript is clearly written, with an appropriate length. The results are novel, convincing, and not oversold. Sufficient methodological details are provided, allowing reproducing the analyses. As the claims are appropriately discussed in the context of a fair treatment of previous literature, they contribute to a better understanding of the evolution of complex red algal-derived plastids (in this context, Figure 1 is of prime importance). Because this manuscript shows how a phylogenetic timeframe can allow testing for different scenarios of symbioses, it will be of interest to researchers in the field, and influence their thinking as well.

I have the following major and minor comments — for example about statistical analyses and methodological investigations — that I hope the authors will consider in a revised, strengthened, future version of their manuscript.

MAJOR POINTS

1. Lines 116-119. We know that taxon sampling is of paramount importance to infer phylogenies, especially for deep cladogenetic events. Therefore, it is important to argue why the pruning of the 733-OTU dataset to obtain the 136-OTU and 63-OTU reduced datasets did not significantly impact the topologies inferred. In other words, what is the effect on the tree inference of the taxonomic diversity, the fast / slow branches, the missing data, and the calibration information (cf. L. 418-420)? With these criteria, is there any trade-off problem to be solved, e.g., keeping a longer branch and/or a superprotein with more missing data provided it brings additional taxonomic representatives or calibration constraints? And what about the possibility of sorting / discarding individual protein alignments according to their average rate of amino acid replacements?

> We thank the reviewer for this comment. The reviewer mentions several factors, including taxon sampling, evolutionary rate variation, and missing data, which often impact on phylogenetic analyses. Indeed, our motivation for investigating both a large (733 OTU) and more tractable (136/63 OTU) datasets was to evaluate the impact of taxon sampling. In the present case, sampling does not appear to have a major impact on our analyses, because the trees recovered from the three datasets were in good agreement and generally matching—in particular for the evolution of algae with red plastids—what could be expected based on previous knowledge, despite differences in taxon sampling and model fit. This result indicates that, fortunately, no difficult trade-offs of the type mentioned by the reviewer seems critical. To be clearer about this, we reworked the descriptions of the trees to also including that of the full dataset and now refer to a recent reference that describes the eukaryotic tree. It now goes as follow: “*The tree based on the full dataset was in good overall agreement with the current consensus of the broad eukaryote phylogeny and classification*²⁶, despite including some highly incomplete and fast-evolving taxa and being derived from the site-homogeneous LG+G+F model. As expected from this model and such a heterogeneous taxon-sampling, the deeper nodes were generally unsupported and we observed cases of long branch attraction, for instance the grouping of *Metamonada*, *Microsporidia*, and *Archamoebae* (Supplementary Fig. 1).” It is however difficult to more fully address the influence of the taxon-sampling since the full dataset cannot be analysed with the same models due to computational demand. Regarding the question of gene evolutionary rate, we modelled across-site rate variation using a gamma distribution in all our analyses. For deep phylogenies, in addition to site rate variation, site and branch compositional heterogeneity (variation of sequence composition across the branches of the tree and/or the sites of the alignment) can interfere with the inference of tree topology and branch length. In the original analysis, we used the CAT+GTR+G model to account for across-site compositional heterogeneity. In the revised manuscript, we have also performed analyses in which we removed the top 25% and 50% of sites that contribute to branch heterogeneity. This was done to evaluate the potential negative impact of compositional biases that could still result in model misspecification in spite of using the best-fitting CAT+GTR+G model. Interestingly, both trees fully replicate the tree derived from the complete alignment. We added the following text to describe this new analysis: “*to help reducing compositional heterogeneity, we performed site stripping of the compositionally most biased sites. The 25% and 50% most compositionally heterogeneous sites were stripped from the 63-OTU alignment, and trees were reconstructed with catgtrg (Supplementary Fig. 6). Both analyses fully confirmed the catgtrg tree recovered from the*

full-length alignment, with only a minor exception in the position of the apusozoan Nutomonas, which moved sister to Discoba in the shortest alignment (Supplementary Fig. 6b)."

2. L195-198: To summarize all estimated node ages, the mean is provided with respect to different calibration priors. However, if these age estimates are not Gaussian-distributed, the mean is not a valid statistical descriptor, and it would be better to use the median with quantiles. These quantiles would also allow fair comparisons among results obtained from different approaches. For example, when it is stated that "The younger ages inferred under the autocorrelated clock model were more apparent when uniform calibrations were applied (mean ages of 904 vs. 947 Ma) and this trend was inverted in the case of skew normal distributions (913 vs. 896 Ma)", there is no statistical background to support this claim because of the partial overlap of credibility intervals.

> We thank the reviewer for this suggestion, this is indeed better. We now provide the median ages throughout.

3. L798. In the calibrations used for dating the eukaryote tree of life, a minimum bound at 1,600 Ma (Paleo/Meso-proterozoic eon) has been set for Rhodophyta. Then, the authors conclude that both the fossil record and molecular clock inferences support a Paleoproterozoic origin of primary plastids, and an early Mesoproterozoic origin of red algae (L. 299-300). The possibility of circularity in these results should however be evaluated. For example, under a leave-one-out procedure, is there any change in the node age estimates if the Rhodophyta calibration is removed? And given the calibrating information constrained on the other nodes, would we again infer early Mesoproterozoic estimates for Rhodophyta? More generally, what is the sensitivity of the molecular clock analyses with respect to the various calibration constraints?

> This is an interesting point. We tested the effect of removing the oldest calibration in our analyses on red algae at 1600 Ma and the inferred ages did not change in essence. In particular, we found consistent ages for the Mesoproterozoic origin of red algae (1377-1207 and 1770-1623 Ma under uncorrelated and autocorrelated clock models, respectively) as well as the Paleoproterozoic origin of primary plastids under the uncorrelated clock model (1964-1860). However, the autocorrelated clock model inferred slightly younger ages for the origin of primary plastids in the Mesoproterozoic (1541-1462 Ma). More generally, we showed that the removal of this fossil does not drastically alter the inferred time ranges, which is relevant given that this fossil is the oldest known eukaryotic fossil. Accordingly, we added the following text to describe these results: "*Finally, the removal of the oldest calibration for the crown-group of red algae, set at 1,600–1,900 Ma²⁴, shifted most (82%) node ages towards present by an average of 127 Ma under the autocorrelated clock model, while the age differences were unappreciable under the uncorrelated clock model (mean of 6 Ma across all nodes). Importantly, however, the 95% HPD intervals remained overlapping between the red algal plastid donor lineage and the origination periods of all lineages with red complex plastids, suggesting that our inferences regarding the rhodoplex hypothesis are robust to varying interpretation of this ancient Proterozoic fossil (Supplementary Fig. 9).*" We think, however, that a general calibration cross-validation strategy (e.g., removing one calibration at a time) would provide limited insights at a very expensive computational cost. Molecular clock studies have shown that such a posteriori validation of calibrations can produce misleading results, and instead that calibrations should be chosen after a careful a priori evaluation (as done in Table 1) and used together (e.g., Warnock et al. 2015 Proc. R. Soc. B <https://doi.org/10.1098/rspb.2014.1013>)

4. The reader is not informed about data availability (alignments, trees, chronograms).

> We thank the reviewer for noticing that we missed to provide a “Data availability” statement, which we corrected accordingly.

MINOR POINTS

1. Lines 127-128: Trees inferred are said to be "in good overall agreement" with current knowledge of the broad eukaryote evolution. Similarly, one tree is "globally consistent" with others (L. 142). These are vague wordings. Actually, which percentage of the total number of nodes is compatible with a given reference topology?

> We rephrased and regrouped the sentences in question in order to tone down our statements and be less general. We also added a recent reference to more directly point to the current consensus we refer to (Burki et al. 2020 Trends in Ecol Evol). This is how it now reads: “*The tree based on the full dataset was in good overall agreement with the current consensus of the broad eukaryote phylogeny and classification²⁶, despite including some highly incomplete and fast-evolving taxa and being derived from the site-homogeneous LG+G+F model. As expected from this model and such a heterogeneous taxon-sampling, the deeper nodes were generally unsupported and we observed cases of long branch attraction, for instance the grouping of Metamonada, Microsporidia, and Archamoebae (Supplementary Fig. 1). The more robust ML-c60 tree derived from the 136-OTU dataset recovered many proposed supergroups with maximal bootstrap support (Supplementary Fig. 2), ...*”. However, we think that giving a number of matching nodes or percentage to compare the trees would not make our point clearer given that the comparison is complicated by the different taxon sampling. Also note that we refer to the corresponding figures so that hopefully the reader can follow our reasoning.

2. L146-147: It is written, "topological differences observed are derived from the chosen evolutionary model and not from the use of Bayesian or ML inference." Why do you expect differences from these two different probability approaches? Please expand on this matter, and/or provide a reference. If you are thinking about prior issues, state it clearly.

> Thank you for pointing this out. We referred to earlier observations (Janoušková, J. et al. A New Lineage of Eukaryotes Illuminates Early Mitochondrial Genome Reduction. *Curr Biology Cb* 27, 3717-3724.e5 (2017)) that showed different placement of *A. twista* depending on the use of ML or Bayesian, but this was not investigated further (in contrast to the present work where we can point to the influence of the model). We now refer to this earlier work in the text.

3. L377, and throughout the text: the authors mention "genes" for some analyses, whereas I assume that "proteins" have been analyzed. Please clarify what is the primary source of information (nucleotides or amino acids).

> The reviewer is correct, we analysed amino acid alignments, to which we occasionally simply refer as “genes”. To avoid confusion, we now explain that “*Throughout this study, amino acid sequences were used for phylogenomic analyses*” (see first sentence of the Methods).

4. L125. As the site heterogeneous CAT+GTR model can also be run without Gamma distribution, the simpler "catgtr" name is rather confusing. Why not using "catgtrg", or more simply keeping CAT+GTR+G?

> We replaced with catgtrg throughout.

5. L441. As a character is a column in a sequence alignment, I assume that the "undetermined characters" are "undetermined / missing character states".

> Corrected as suggested.

6. Even if the manuscript is centered on time-calibrated phylogenies, it might be worth providing the reader with a supplementary figure showing the highest-likelihood or maximum probability tree with branch lengths. This would allow appreciating the extent of evolutionary rate variations among clades.

> Thank you for this suggestion, the tree with branch lengths can now be found in the Supplementary Information (Fig. S4). Fig. S4 is mentioned in the main text as well as in the legend of the time-calibrated tree.

7. Figure 1 is very important to understand the scientific context of this manuscript. For this reason, it would help to place the primary endosymbiosis event leading to primary plastids in photosynthetic eukaryotes.

> We have done as suggested, and thank you for noting the importance of this figure.

Reviewers' Comments:

Reviewer #1:

Remarks to the Author:

The authors addressed my comments adequately and I support the publication of the article in Nature Communications. I have one comment though: "we observed cases of long branch attraction, for instance 136 the grouping of Metamonada, Microsporidia, and Archamoebae"  this strange grouping may not be solely to do with LBA. There are published papers that describe genes of parabasalid origins in Entamoeba.

Congrats.

-Eunsoo Kim

Reviewer #2:

Remarks to the Author:

My original review contained only suggested revisions in the first place, and I am satisfied the authors have responded, at a responsible level, to the issues and concerns raised by all three reviewers. In particular, I appreciate their effort to address the question of whether EGT could be confounding both the overall topology of the tree, as well as the inferred internode lengths that are key to molecular clock calibrations. They went beyond my request to discuss these issues, instead carrying out a global analysis to look for conflicting sister-relationships in phylogenetic signals that could be impacting the positions of various red-derived plastid lineages. Given that few major studies have even taken the issue seriously, much less tried to address it, the authors are to be commended for the effort. That said, their "assessment of the signal for close relationships in single gene trees, averaged over bootstrap replicates for the entire marker gene set" probably itself could be confounded by the same issues they are trying to address.

For example, let's take a worst-case scenario; that is, there is virtually no consistent genealogical phylogenetic signal present in sequences for deeper branches of the tree, and EGT itself actually accounts for the dominant signal for key regions of the tree's structure. Clearly, in this case, there could be no mixed signals in a test for EGT as a complication of the analysis, because there is no strong "host cell" signal in the first place. Although it is easy to dismiss such an "all or nothing" scenario, the absence of strong phylogenetic signal for ancient events is well known, and the fact that strong conflicting signals do not show up in a global analysis is perhaps no more surprising than the fact that these branches are difficult to recover under any circumstances. Moreover, this is but one potential complication. The antiquity of the transfers in question combined with subsequent extinctions and phylogenetic radiations are not trivial issues to consider in such an analysis. And if serial movement of plastids really occurred, then are conflicting individual sister-relationships even to be expected, especially when averaging across the entire tree? It is not clear to me that this is a valid expectation when dealing with in multi-chimeric genomes containing sequences from several distantly related organisms. In other words, the experimental design may be inappropriate to the question addressed, and the methodology could prove circular if EGT is an important driver of key parts of the consensus tree topology.

My original comments were not intended to prompt the authors to quickly come up with a global analysis, particularly one that could be as fraught with biases as the consensus phylogenetic analysis itself. Rather, I was hoping this committed and talented group of researchers might consider further research that developed explicit tests of specific hypotheses of red-derived plastid evolution, and perhaps go on to address the issue of EGT within more restricted and controllable research frameworks. That way, the methodologies could be fully vetted in their own right, both by the researchers and by qualified reviewers.

Although I would leave a decision on whether include this analysis to the authors and editors, I am concerned that publishing it as an afterthought in this study could actually have a negative impact on the very issue the authors want to address. As noted in my original comments, many researchers in this field simply don't want to deal with some of the thornier problems underlying broad scale phylogenomic investigations. Do these authors really want to provide support for that mentality without first taking the time to develop the most rigorous approach possible toward addressing the potential impacts of EGT?

Reviewer #3:

Remarks to the Author:

I reviewed a previous version of this manuscript and I thank the authors for constructively considering the comments of the reviewers. I enjoyed reading this new version of the manuscript as the authors have done a very good job in addressing the various points. This revision is significantly improved from the original version, and my comments represent cause for (at most) minor revisions. [Line numbering corresponds to the PDF with tracking changes.]

1. Line 37 (Abstract). "phylogenomic dataset". You might provide the maximum number of taxa and markers / sites here analyzed.
2. L112. Do you mean "genes", "protein-coding genes", "markers" or "proteins"?
3. L133-139. For the broad readership of Nature Communications, briefly clarify why the so-called site-homogeneous LG+G+F model is "expected" to yield generally unsupported deeper nodes.
4. L163-164. "The catgtrg topology was also accepted by ML in an Approximately Unbiased test". Strictly speaking, the AU test "does not reject" an alternative topology, hence "accepting" it.
5. L203-204. Bootstrap support [...] favoured the "branching of the" lineage that would be expected based on prior knowledge of species relationships.
6. L263 (and throughout the text). A note of caution about "young" or "younger" ages: "young" may either refer to "close to present" (a common biological acceptation), or to "close to the origin of Earth" (cf. a geological context).
7. L422-423. 'The serial transfers all took place in a relatively "short" time window between 650 and 1,079 million years from the initial secondary endosymbiosis'. Actually, a time window of 429 million years is not so "short". It depends upon the scientific context (geology, biology, phylogeny, astrophysics ...).
8. L512-514. "In cases of very recent gene duplications, characterised by two or more paralogs of the same taxon, those with the longest branches were removed." I assume this is a strategy to reduce long branch attraction phenomena.

REVIEWERS' COMMENTS

Reviewer #1 (Remarks to the Author):

The authors addressed my comments adequately and I support the publication of the article in Nature Communications. I have one comment though: "we observed cases of long branch attraction, for instance 136 the grouping of Metamonada, Microsporidia, and Archamoebae"  this strange grouping may not be solely to do with LBA. There are published papers that describe genes of parabasalid origins in Entamoeba.
Congrats.

-Eunsoo Kim

> We thank Dr. Kim for her valuable comments in the entire revision process. As for the grouping of Metamonada, Microsporidia, and Archamoebae, we now mention LGT as a further phenomenon causing groupings between only distantly related groups: "However, lateral gene transfers among these groups may also account for their grouping (REF)."

Reviewer #2 (Remarks to the Author):

My original review contained only suggested revisions in the first place, and I am satisfied the authors have responded, at a responsible level, to the issues and concerns raised by all three reviewers. In particular, I appreciate their effort to address the question of whether EGT could be confounding both the overall topology of the tree, as well as the inferred internode lengths that are key to molecular clock calibrations. They went beyond my request to discuss these issues, instead carrying out a global analysis to look for conflicting sister-relationships in phylogenetic signals that could be impacting the positions of various red-derived plastid lineages. Given that few major studies have even taken the issue seriously, much less tried to address it, the authors are to be commended for the effort. That said, their "assessment of the signal for close relationships in single gene trees, averaged over bootstrap replicates for the entire marker gene set" probably itself could be confounded by the same issues they are trying to address.

For example, let's take a worst-case scenario; that is, there is virtually no consistent genealogical phylogenetic signal present in sequences for deeper branches of the tree, and EGT itself actually accounts for the dominant signal for key regions of the tree's structure. Clearly, in this case, there could be no mixed signals in a test for EGT as a complication of the analysis, because there is no strong "host cell" signal in the first place. Although it is easy to dismiss such an "all or nothing" scenario, the absence of strong phylogenetic signal for ancient events is well known, and the fact that strong conflicting signals do not show up in a global analysis is perhaps no more surprising than the fact that these branches are difficult to recover under any circumstances. Moreover, this is but one potential complication. The antiquity of the transfers in question combined with subsequent extinctions and phylogenetic radiations are not trivial issues to consider in such an analysis. And if serial movement of plastids really occurred, then are conflicting individual sister-relationships even to be expected, especially when averaging across the entire tree? It is not clear to me that this is a valid expectation when dealing with in multi-

chimeric genomes containing sequences from several distantly related organisms. In other words, the experimental design may be inappropriate to the question addressed, and the methodology could prove circular if EGT is an important driver of key parts of the consensus tree topology.

My original comments were not intended to prompt the authors to quickly come up with a global analysis, particularly one that could be as fraught with biases as the consensus phylogenetic analysis itself. Rather, I was hoping this committed and talented group of researchers might consider further research that developed explicit tests of specific hypotheses of red-derived plastid evolution, and perhaps go on to address the issue of EGT within more restricted and controllable research frameworks. That way, the methodologies could be fully vetted in their own right, both by the researchers and by qualified reviewers.

Although I would leave a decision on whether include this analysis to the authors and editors, I am concerned that publishing it as an afterthought in this study could actually have a negative impact on the very issue the authors want to address. As noted in my original comments, many researchers in this field simply don't want to deal with some of the thornier problems underlying broad scale phylogenomic investigations. Do these authors really want to provide support for that mentality without first taking the time to develop the most rigorous approach possible toward addressing the potential impacts of EGT?

> We are grateful to Dr. Stiller as his thorough comments about EGT allowed us to provide a more comprehensive version of our manuscript. Despite the concerns noted, our preference is to keep the new analysis in the text as we think it provides a fresh and reliable perspective on the potential issues of EGT wrongly influencing the relationship. Indeed, to our knowledge the systematic check we performed at the single-gene level has not been done before on similar datasets, and since we have relatively specific hypotheses about potential donors and recipients we can see that non-photosynthetic clades correctly appear related to their photosynthetic clades in most cases. In addition, this analysis addresses an earlier issue raised by R1. We, however, now acknowledge the potential caveats by noting that "since this analysis relies on gene tree conflict to identify EGTs, a lineage in which most or all marker genes have been replaced by EGT might not be detected."

Reviewer #3 (Remarks to the Author):

I reviewed a previous version of this manuscript and I thank the authors for constructively considering the comments of the reviewers. I enjoyed reading this new version of the manuscript as the authors have done a very good job in addressing the various points. This revision is significantly improved from the original version, and my comments represent cause for (at most) minor revisions. [Line numbering corresponds to the PDF with tracking changes.]

> We would like to thank Reviewer 3 for the positive feedback and helpful comments throughout the revision process. Please find below our final comments and corrections.

1. Line 37 (Abstract). "phylogenomic dataset". You might provide the maximum number of taxa and markers / sites here analyzed.

> As we analysed several datasets, which all were of importance for the inference of the final tree that has been used for molecular clock analyses, we prefer to not provide these numbers for one single dataset in the Abstract.

2. L112. Do you mean "genes", "protein-coding genes", "markers" or "proteins"?

> We added "protein-coding genes" to avoid confusion.

3. L133-139. For the broad readership of Nature Communications, briefly clarify why the so-called site-homogeneous LG+G+F model is "expected" to yield generally unsupported deeper nodes.

> We now briefly mention that "site-homogeneous models do not capture well site-specific amino acid preference and as a result can cause systematic errors in phylogenetic estimation (REF)."

4. L163-164. "The catgtrg topology was also accepted by ML in an Approximately Unbiased test". Strictly speaking, the AU test "does not reject" an alternative topology, hence "accepting" it.

> Thank you, we modified the text accordingly.

5. L203-204. Bootstrap support [...] favoured the "branching of the" lineage that would be expected based on prior knowledge of species relationships.

> Corrected.

6. L263 (and throughout the text). A note of caution about "young" or "younger" ages: "young" may either refer to "close to present" (a common biological acceptation), or to "close to the origin of Earth" (cf. a geological context).

> Throughout the manuscript younger refers to "close to present". To avoid confusion, we now specify this where mentioned first time.

7. L422-423. 'The serial transfers all took place in a relatively "short" time window between 650 and 1,079 million years from the initial secondary endosymbiosis'. Actually, a time window of 429 million years is not so "short". It depends upon the scientific context (geology, biology, phylogeny, astrophysics ...).

> We agree with the reviewer that this is highly dependent on the context and therefore stated "relatively short" instead of "short".

8. L512-514. "In cases of very recent gene duplications, characterised by two or more paralogs of the same taxon, those with the longest branches were removed." I assume this is a strategy to reduce long branch attraction phenomena.

> Correct. A corresponding statement has been added.